# CGSA: Class-Guided Slot-Aware Adaptation for Source-Free Object Detection

**Boyang Dai**[1*], **Zeng Fan**[2*], **Zihao Qi**[3], **Meng Lou**[1], **Yizhou Yu**[1,4†]

[1]School of Computing and Data Science, The University of Hong Kong
[2]Department of Electrical & Computer Engineering, National University of Singapore
[3]School of Information and Electronics, Beijing Institute of Technology
[4]Center for Embodied AI and Computer Vision, Shenzhen Loop Area Institute
{ boyangdai@connect.hku.hk, e1521148@u.nus.edu, 3120255601@bit.edu.cn,
loumeng@connect.hku.hk, yizhouy@acm.org }

## Abstract

Source-Free Domain Adaptive Object Detection (SF-DAOD) aims to adapt a detector trained on a labeled source domain to an unlabeled target domain without retaining any source data. Despite recent progress, most popular approaches focus on tuning pseudo-label thresholds or refining the teacher-student framework, while overlooking object-level structural cues within cross-domain data. In this work, we present **CGSA**, the first framework that brings Object-Centric Learning (OCL) into SF-DAOD by integrating slot-aware adaptation into the DETR-based detector. Specifically, our approach integrates a **Hierarchical Slot Awareness** (HSA) module into the detector to progressively disentangle images into slot representations that act as visual priors. These slots are then guided toward class semantics via a **Class-Guided Slot Contrast** (CGSC) module, maintaining semantic consistency and prompting domain-invariant adaptation. Extensive experiments on multiple cross-domain datasets demonstrate that our approach outperforms previous SF-DAOD methods, with theoretical derivations and experimental analysis further demonstrating the effectiveness of the proposed components and the framework, thereby indicating the promise of object-centric design in privacy-sensitive adaptation scenarios. Code is released at https://github.com/Michael-McQueen/CGSA.

## 1 Introduction

Real-world deployment of object detectors (Ren et al., 2015; Redmon et al., 2016; Carion et al., 2020) inevitably encounters domain shifts, such as changes in weather, camera, or scene layout, which cause dramatic performance drops when the model is trained on a different, labeled source domain. Unsupervised domain-adaptive object detection (DAOD) addresses this gap by adapting a detector to an unlabeled target domain while still having full access to the source data. Existing DAOD methods predominantly align global or instance features through adversarial learning (Chen et al., 2018), query-token matching (Huang et al., 2022), or teacher-student consistency (Li et al., 2022b). Although these techniques yield solid

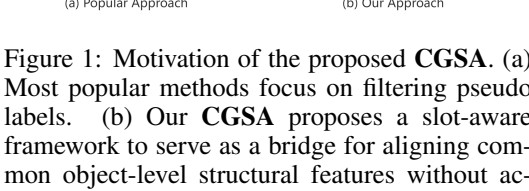

Figure 1: Motivation of the proposed **CGSA**. (a) Most popular methods focus on filtering pseudo labels. (b) Our **CGSA** proposes a slot-aware framework to serve as a bridge for aligning common object-level structural features without accessing source data.

---

*Equal contribution.
†Corresponding author.

gains, they presuppose that source images remain available during adaptation, an assumption often violated by privacy regulations or proprietary constraints.

In Source-Free DAOD (SF-DAOD), this dependency is relaxed because the adaptation phase receives only a detector pretrained on the source domain and the target images themselves. Because source data are unavailable and target labels are absent, SF-DAOD typically adopts a *teacher–student* paradigm (Tarvainen & Valpola, 2017): the *teacher* predicts on target images, filtered predictions form pseudo labels, and the *student* is trained on these signals. Consequently, current approaches focused on refining the framework or the selection scheme, such as confidence filtering (Li et al., 2021), periodic smoothing (Liu et al., 2023), and energy-based pruning (Chu et al., 2023). Despite recent progress, popular methods largely ignore the object-level structural regularities that persist across source and target domains. As a result, the source-trained detector is relegated to a mere pseudo-label oracle, and its rich internal representations remain under-exploited.

In response to the above challenges, Object-Centric Learning (OCL) offers a solution. Slot Attention (Locatello et al., 2020), a representative OCL technique, decomposes a scene into a small set of latent "slots", each expected to bind to one object and thus inherently separate foreground entities from background context. Slot-based models have shown impressive transferability in many vision tasks (Seitzer et al., 2022; Zhou et al., 2022; Jiang et al., 2023), yet have never been explored for SF-DAOD. Given that DETR-based detectors (Carion et al., 2020) already rely on object queries, embedding slot-aware priors into the query space is a natural but unexplored direction.

Consequently, we propose **CGSA**, the first framework to merge OCL with source-free adaptive detection. Figure 1 shows our motivation. **CGSA** acquires slot-based structural visual priors through a progressive visual decomposition mechanism, and guides the resulting slots with class prototypes toward domain-invariant representations via contrastive learning that evolves as the detector improves. Together, by explicitly extracting, guiding, and leveraging these shared structural cues, our framework allows the pretrained model to contribute more directly and effectively during adaptation.

The main contributions are summarized as follows:

(i) To the best of our knowledge, we are the first to introduce OCL into the SF-DAOD problem, establishing a new slot-aware adaptation framework **CGSA**.

(ii) We devise two complementary modules, namely **Hierarchical Slot Awareness** (HSA), **Class-Guided Slot Contrast** (CGSC), which collectively provide structural visual priors and semantic guidance to exploit domain-invariant slot awareness under the source-free setting. We further provide a theoretical generalization analysis supporting our method.

(iii) Extensive experiments on multiple datasets, complemented by ablation studies and visualization analyses, demonstrate the effectiveness of the proposed components and framework.

## 2 RELATED WORKS

**Unsupervised Domain Adaptive Object Detection** (DAOD) aims to transfer a detector trained on a labeled source domain to an unlabeled target domain. Early works extend feature-level adversarial learning from classification to detection. For example, DA Faster R-CNN (Chen et al., 2018) aligns image-level and instance-level features with two discriminators. With the advent of transformer detectors, query-aware alignment becomes feasible. MTTrans (Yu et al., 2022) and AQT (Huang et al., 2022) match cross-domain tokens or queries via the attention mechanism. DATR (Chen et al., 2025) further couples alignment across classes. The third technical route introduces consistency or self-training signals. AT (Li et al., 2022b) exploits momentum teacher, MRT (Zhao et al., 2023) adds masked reconstruction, and CAT (Kennerley et al., 2024) regularizes category centroids across domains. Despite steady progress, these approaches presuppose full access to source images throughout adaptation, an assumption that is often violated when data sharing is restricted by privacy or licensing.

**Source-Free Domain Adaptive Object Detection** (SF-DAOD) removes that dependency: only a source-trained model and unlabeled target images are available. Since a *teacher–student* paradigm is required to obtain pseudo labels for supervision, current solutions mainly optimized the framework or the selection scheme. SFOD and SFOD-M (Li et al., 2021) select high-confidence boxes from the teacher; LODS (Li et al., 2022a) and PETS (Liu et al., 2023) refine labels by objectness

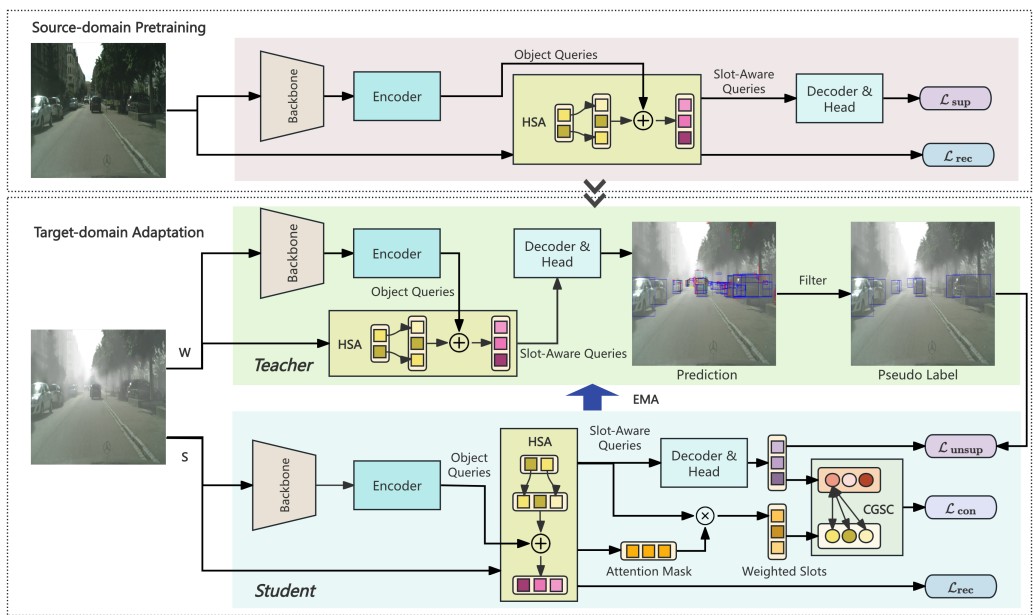

Figure 2: Framework of **CGSA**. It consists of two stages: source-domain pretraining and target-domain adaptation, which work collaboratively to improve SF-DAOD performance.

or temporal smoothing; A$^2$SFOD (Chu et al., 2023) and TITAN (Ashraf & Bashir, 2025) partition the target domain into source-similar and source-dissimilar subsets according to variance, thereby recasting the source-free setting as a traditional DAOD that can be addressed with existing techniques. However, these methods mostly concentrate on refining pseudo-labels while neglecting the object-level structural information already embedded in the detector's representations. Addressing the limitation requires a mechanism that can explicitly extract individual objects without access to the source data, and our work is designed to fill this gap.

**Object-Centric Learning** (OCL) decomposes scenes into discrete entity representations, enabling reasoning that is invariant to background shifts. Slot Attention (Locatello et al., 2020) iteratively binds "slots" to latent objects via attention. In vision tasks, slot-based models have shown strong transferability in real-world segmentation (Seitzer et al., 2022), video prediction (Zhou et al., 2022), generative modeling (Jiang et al., 2023), and even robotics (Rezazadeh et al., 2024), yet their potential for domain adaptation remains largely unexplored, especially in detection, where DETR-based decoders already employ object queries but lack structural priors. Therefore, we bridge this gap by proposing a novel slot-aware framework that unifies the object-centric paradigm for SF-DAOD.

## 3 METHOD

### 3.1 OVERVIEW

Under the source-free setting, *object-centric disentanglement* provides a promising route, which separates potential targets from the surrounding scene. Once objects are isolated, object-level knowledge can be transferred directly, enabling implicit cross-domain adaptation without any source samples. Motivated by this observation, we propose **CGSA**, a source-free object detection framework with class-guided supervision for fine-grained visual priors, as illustrated in Figure 2. Pseudocode for the core operations is provided in the Appendix E.

An input image is first processed by the backbone and query encoder to obtain object queries. In parallel, the same image passes through **Hierarchical Slot Awareness** (HSA), which decomposes it into a coarse-to-fine set of $n^2$ slots. After projection, these slots fuse with the queries to form slot-aware queries carrying object-level visual prior into the decoder.

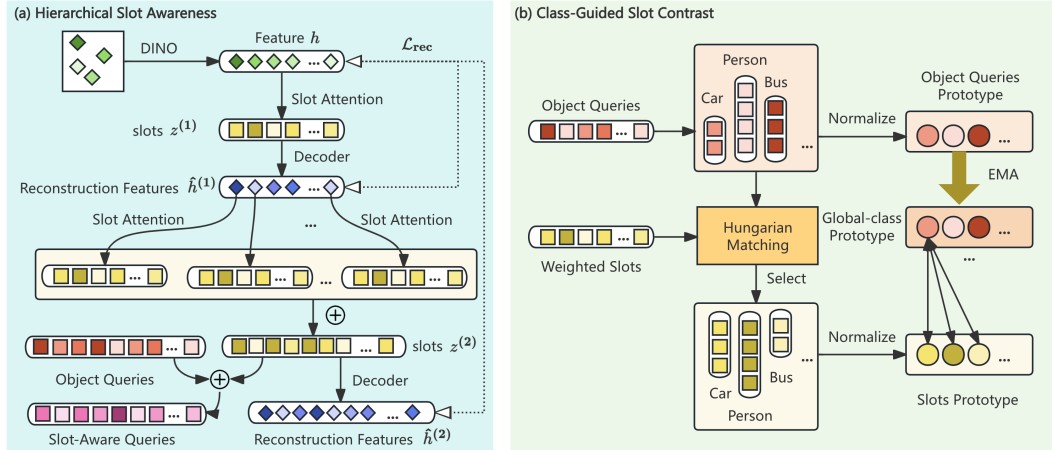

Figure 3: (a) Pipeline of the proposed **HSA** module. Through the hierarchical design, the features are first decomposed into coarse-to-fine slots. After projection, these slots are concatenated with the object queries to form slot-aware queries, thereby providing object-level structural priors. (b) Pipeline of the proposed **CGSC** module. By maintaining the global class prototype and assigning pseudo labels to weighted slots for class attributes, contrastive learning is used to implicitly supervise and guide the slots to focus on domain-invariant yet class-relevant object features.

**Source-domain pretraining.** The detector is trained with standard detection supervision while **HSA** adds a reconstruction objective, and together they provide strong structural priors for subsequent adaptation.

**Target-domain adaptation.** A *teacher-student* configuration is adopted, both initialized from the source-domain pretrained model. The *student* repeats the **HSA** path on target images, and the resulting attention mask weights the queries to produce weighted slots that are contrasted with dynamic class prototypes by the **Class-Guided Slot Contrast** (CGSC) module. The *teacher* generates predictions on the same image, selecting reliable pseudo labels for the *student* based on a threshold. The weights of *teacher* are updated by the exponential moving average (EMA) of the *student*.

## 3.2 HIERARCHICAL SLOT AWARENESS

We introduce a **Hierarchical Slot Awareness** (HSA) module, shown in Figure 3(a), that brings OCL priors into DETR-based detectors. **HSA** structurally disentangles an image into object-level slots in a coarse-to-fine manner and embeds these regional visual prior representations into the object queries. In this way, we realize object-level cross-domain adaptation in a principled and stable fashion. Ablation studies and visualization results are provided in the Appendix G.

### 3.2.1 DECOMPOSING VISUAL PRIORS

Grounded in evidence that human vision automatically decomposes sensory input into a small set of parts under strong internal visual priors (Biederman, 1987), we therefore cast visual prior decomposition as an iterative inference and refinement process. Slot Attention (Locatello et al., 2020) meets the conditions, whose core idea is to iteratively aggregate information from the input features into a set of slots. In each iteration, the $k$-th slot computes an attention weight with an input feature $h_i$, updates itself, and after several iterations (typically 3–5) each slot $z_k$ becomes a latent representation of an individual object. The process is expressed as follows:

$$z_k^{(t+1)} = \text{GRU}\left( \sum_{i=1}^{N} \text{softmax}_i\left( \frac{(z_k^{(t)} W_q)(h_i W_k)^\top}{\sqrt{d}} \right) h_i W_v, \ z_k^{(t)} \right), i \in \{1, \dots, N\}, \ k \in \{1, \dots, n\}.$$

(1)

where $h_i \in \mathbb{R}^{1 \times d}$ is the $i$-th input feature, $z_k^{(t)} \in \mathbb{R}^{1 \times d}$ denotes the $k$-th slot state at iteration $t$, $W_q$, $W_k$, and $W_v \in \mathbb{R}^{d \times d}$ are shared linear projection matrices. For brevity, we omit the image index

in the following notation. $N$ denotes the number of token features per image, and $n$ denotes the corresponding number of slots.

Given that decomposing process is unsupervised, to bind these slots to spatial support and regularize discovery, we decode slots with a lightweight spatial-broadcast MLP (Seitzer et al., 2022), obtaining per-slot reconstructions $\hat{h}_k \in \mathbb{R}^{N \times d}$ and mask logits $\alpha_k \in \mathbb{R}^{N \times 1}$. Competition among regions is enforced by a softmax across slots, $m_k = \mathrm{softmax}_k(\alpha_k)$, and the optimized way is to reconstruct, defined as below.

$$\hat{h} = \sum_{k=1}^{n} \hat{h}_k \odot m_k, \qquad \mathcal{L}_{\mathrm{rec}} = \left\| \hat{h} - h \right\|_2^2. \tag{2}$$

### 3.2.2 HIERARCHICAL ARCHITECTURE

Traditional slot-based decomposition on real-world datasets limits the number of slots (typically $\leq 10$) to ensure convergence and avoid collapse (Seitzer et al., 2022). To break this constraint while retaining stability, we take inspiration from evidence that human vision performs a progressive decomposition of scenes into parts and relations (Schyns & Oliva, 1994), first establishing coarse structural descriptions and then refining detail. Therefore, we adopt a hierarchical design in which a first-stage decoupling extracts coarse, region-level visual priors and a second-stage decoupling refines them into finer-grained priors. Concretely, we first obtain coarse-grained priors, denoted $z^{(1)} \in \mathbb{R}^{n \times d}$ and $\hat{h}^{(1)} \in \mathbb{R}^{N \times d}$, as described in Section 3.2.1. We then treat the latter as the raw features for the next level, apply the same refinement rule as in Eq 1, and concatenate the resulting representations to obtain $z^{(2)} \in \mathbb{R}^{n^2 \times d}$.

A mirrored decoder predicts $\{\hat{h}_k^{(2)}, \alpha_k^{(2)}\}_{k=1}^{n^2}$ with masks $m_k^{(2)} = \mathrm{softmax}_k(\alpha_k^{(2)})$, yielding the reconstruction $\hat{h}^{(2)} = \sum_{k=1}^{n^2} \hat{h}_k^{(2)} \odot m_k^{(2)}$. We supervise awareness formation at both levels with the total objective:

$$\mathcal{L}_{\mathrm{rec}} = \mathcal{L}_{\mathrm{rec}}^{(1)} + \mathcal{L}_{\mathrm{rec}}^{(2)} = \left\| \hat{h}^{(1)} - h \right\|_2^2 + \left\| \hat{h}^{(2)} - h \right\|_2^2. \tag{3}$$

In practice, we set $n=5$, resulting in $n^2=25$ fine slots, which retains training stability while providing finer region representations and adequate granularity. Ablation studies and visualization results are provided in the Appendix G.

### 3.2.3 SLOT-AWARE QUERIES

The slots encode structural cues of potential objects or regions. To infuse this information into the decoder, we first project the slot representations through a linear mapper $f_{\mathrm{map}}$ so that their dimension matches that of the object queries. We then add them to the original queries $Q_{\mathrm{obj}} \in \mathbb{R}^{M \times d_q}$ to obtain the slot-aware queries as final decoder input:

$$\tilde{z}^{(2)} = f_{\mathrm{map}}\left(z^{(2)}\right) \in \mathbb{R}^{M \times d_q}, \quad Q_{\mathrm{aware}} = Q_{\mathrm{obj}} + \tilde{z}^{(2)}. \tag{4}$$

This fusion endows slot-aware queries with object-level structural priors, enabling more domain-invariant localization, while the original queries can be regarded as the weight distribution of the visual priors.

### 3.3 CLASS-GUIDED SLOT CONTRAST

Existing slot-aware mechanism receives no explicit semantic constraints, so some slot vectors $z_k$ tend to absorb domain-specific background noise. To steer the slots toward more domain-invariant yet class-relevant object features, we introduce **Class-Guided Slot Contrast** (CGSC), shown in Figure 3(b). **CGSC** maintains class prototypes online and imposes a contrastive loss, thus providing implicit supervision to guide the slots towards their potential class attributes.

### 3.3.1 CLASS-PROTOTYPE MEMORY

Given the $M$ decoder queries $\{q_i\}_{i=1}^M$ and their predicted classes $\hat{c}_i \in \{1, \ldots, C\}$, we compute a temporary object queries prototype $\bar{p}_c$ for each class by averaging queries of the same class.

$$\bar{p}_c = \frac{1}{|\mathcal{Q}_c|} \sum_{q_i \in \mathcal{Q}_c} q_i, \quad \mathcal{Q}_c = \{q_i \mid \hat{c}_i = c\}. \tag{5}$$

The global-class prototype memory $P_c$ is then updated with an exponential moving average (EMA):

$$P_c \leftarrow \beta P_c + (1 - \beta)\bar{p}_c, \quad \beta \in [0, 1). \tag{6}$$

### 3.3.2 WEIGHTED SLOT CONSTRUCTION

With $h$ denoting the original features and $m_k^{(2)}$ from Section 3.2.2 as the normalized attention mask, the weighted slot $\tilde{z}_k$ is defined as:

$$\tilde{z}_k = \sum_{i=1}^N m_{k,i}^{(2)} \cdot h_i. \tag{7}$$

Because $m_k^{(2)}$ measures the visual evidence assigned to each feature, this aggregation suppresses slots dominated by background regions and keeps $\tilde{z}_k$ in the same representation space as the decoder queries, generating a reliable measure of similarity to class prototypes.

### 3.3.3 PROTOTYPE-SLOT MATCHING AND CONTRASTING

We first measure the cosine similarity between each weighted slot $\tilde{z}_k$ and every decoder query $q_i$ to form the matrix $S_{ki} = \phi(\tilde{z}_k, q_i)$. Applying the Hungarian algorithm (Kuhn, 1955) to $S$ yields a one-to-one assignment $\pi(k)$ that pairs slot $k$ with query $q_{\pi(k)}$. Because each query carries a predicted class label $\hat{c}_{\pi(k)}$, this assignment implicitly labels the slot as $\hat{c}_k = \hat{c}_{\pi(k)}$.

Next, all weighted slots with the same label are averaged to obtain the slots prototype $\bar{z}_c$:

$$\bar{z}_c = \frac{1}{|\mathcal{Z}_c|} \sum_{k \in \mathcal{Z}_c} \tilde{z}_k, \quad \mathcal{Z}_c = \{k \mid \hat{c}_k = c\}. \tag{8}$$

Finally, we compute a contrastive loss (Oord et al., 2018; Chen et al., 2020) between the slot prototypes $\bar{z}_c$ and the global-class prototypes $P_c$ as follows:

$$\mathcal{L}_{\text{con}} = -\frac{1}{C} \sum_{c=1}^C \log \frac{\exp(\phi(P_c, \bar{z}_c)/\tau)}{\displaystyle\sum_{c'=1}^C \exp(\phi(P_c, \bar{z}_{c'})/\tau)}, \tag{9}$$

where $\tau$ is a temperature hyperparameter and $\phi(\cdot, \cdot)$ denotes cosine similarity. This loss pulls each weighted slot toward its matched class prototype while pushing it away from prototypes of other classes, driving the slots to capture class-specific, domain-invariant semantics.

Through the combined forces of class guidance and contrastive regulation, **CGSC** enables **HSA** to disentangle reliable object representations without access to source images. Consequently, the detector achieves fine-grained, class-relevant, and slot-aware adaptation to the target domain.

### 3.4 TOTAL ADAPTATION OBJECTIVE

Given the filtered pseudo labels $\hat{\mathcal{Y}}_\tau = \{(\hat{b}_j, \hat{c}_j) \mid \hat{p}_j \geq \tau(s)\}$, where $\hat{p}_j$ is the teacher's confidence and $\tau(s)$ is the threshold (details in Appendix C), the student's unsupervised detection objective consists of a classification term and a box-regression term, where we use *Focal Loss* (Lin et al., 2017) for classification and a combination of $\ell_1$ and *GIoU loss* (Rezatofighi et al., 2019) for box regression.

$$\mathcal{L}_{\text{unsup}} = \frac{1}{|\hat{\mathcal{Y}}_\tau|} \sum_{(\hat{b}_j, \hat{c}_j) \in \hat{\mathcal{Y}}_\tau} \left[ \mathcal{L}_{\text{cls}}(f_{\text{cls}}(x_t), \hat{c}_j) + \mathcal{L}_{\text{box}}(f_{\text{box}}(x_t), \hat{b}_j) \right]. \tag{10}$$

Then the complete loss that drives the *student* network combines the unsupervised detection loss with the reconstruction loss $\mathcal{L}_{\text{rec}}$ and the contrastive loss $\mathcal{L}_{\text{con}}$:

$$\mathcal{L}_{\text{total}} = \mathcal{L}_{\text{unsup}} + \lambda_{\text{con}} \mathcal{L}_{\text{con}} + \lambda_{\text{rec}} \mathcal{L}_{\text{rec}}. \qquad (11)$$

### 3.5 GENERALIZATION ANALYSIS

We aim to theoretically characterize how our components interact with a DETR-based detector to enable reliable source-free adaptation on the target domain. In brief, our modules (i) contract domain-specific background variance, (ii) enlarge cosine inter-class margins, and (iii) propagate the margin gain through the decoder. Under standard regularity conditions detailed in the Appendix A, these effects combine into a *risk descent* bound for the target objective, with full statements and proofs deferred to the Appendix A.

**Theorem:** *Risk descent on the target domain.*

Let $\mathcal{R}_T(\theta)$ be the target objective in Eqs. 10–11. For the iterate $\theta_t$ produced by minimizing the target objective $\mathcal{R}_T(\theta)$, there exist constants $c_1, c_2 > 0$ such that

$$\mathbb{E}\,\mathcal{R}_T(\theta_{t+1}) \leq \mathbb{E}\,\mathcal{R}_T(\theta_t) - c_1 \Delta_t + c_2 (\epsilon_{\text{rec}} + \sigma^2), \qquad (12)$$

where $\Delta_t$ is the cosine-margin gain induced by prototype–slot InfoNCE (Eq. 9) and propagated by slot-aware query fusion (Eq. 4), while $\epsilon_{\text{rec}}$ and $\sigma^2$ stem from reconstruction consistency (Eq. 3) and bounded background.

The margin term guarantees a monotone decrease in the classification component. The residual is controlled by reconstruction quality and target background noise. Under a mild contraction condition, the induced one-dimensional iterate converges, and the expected target risk decreases monotonically.

## 4 EXPERIMENTS

Table 1: Adaptation from Small-Scale to Large-Scale Dataset: Cityscapes $\rightarrow$ BDD100K. "SF" refers to the source-free setting. "FRCNN" denotes variants based on the Faster R-CNN family; "DETR" denotes variants based on the DETR family. "Oracle" denotes an upper bound obtained by training on the target domain with ground-truth annotations.

| Method | SF | Base | Person | Rider | Car | Truck | Bus | Mcycle | Bicycle | mAP |
|---|---|---|---|---|---|---|---|---|---|---|
| Oracle | - | DETR | 68.2 | 53.2 | 84.0 | 67.7 | 67.9 | 54.1 | 54.7 | 64.3 |
| DA-Faster(Chen et al., 2018) | ✗ | FRCNN | 28.9 | 27.4 | 44.2 | 19.1 | 18.0 | 14.2 | 22.4 | 24.9 |
| SFA(Wang et al., 2021) | ✗ | DETR | 40.2 | 27.6 | 57.5 | 19.1 | 23.4 | 15.4 | 19.2 | 28.9 |
| AQT(Huang et al., 2022) | ✗ | DETR | 38.2 | 33.0 | 58.4 | 17.3 | 18.4 | 16.9 | 23.5 | 29.4 |
| MRT(Zhao et al., 2023) | ✗ | DETR | 48.4 | 30.9 | 63.7 | 24.7 | 25.5 | 20.2 | 22.6 | 33.7 |
| DATR(Chen et al., 2025) | ✗ | DETR | 58.5 | 42.8 | 73.4 | 26.9 | 39.9 | 24.2 | 37.3 | 43.3 |
| SFOD(Li et al., 2021) | ✓ | FRCNN | 31.0 | 32.4 | 48.8 | 20.4 | 21.3 | 15.0 | 24.3 | 27.6 |
| SFOD-M(Li et al., 2021) | ✓ | FRCNN | 32.4 | 32.6 | 50.4 | 20.6 | 23.4 | 18.9 | 25.0 | 29.0 |
| PETS(Liu et al., 2023) | ✓ | FRCNN | 42.6 | 34.5 | 62.4 | 19.3 | 16.9 | 17.0 | 26.3 | 31.3 |
| A²SFOD(Chu et al., 2023) | ✓ | FRCNN | 33.2 | 36.3 | 50.2 | 26.6 | 24.4 | 22.5 | 28.2 | 31.6 |
| TITAN(Ashraf & Bashir, 2025) | ✓ | DETR | 49.9 | 35.6 | 65.7 | 24.6 | 35.9 | 31.5 | 29.2 | 38.3 |
| **CGSA(Ours)** | ✓ | DETR | **60.0** | **47.6** | **75.0** | **48.4** | **53.1** | **43.5** | **43.4** | **53.0** |

### 4.1 DATASET

We conducted experiments on five widely used object detection datasets to evaluate the performance of our proposed method. **(1) Cityscapes** (Cordts et al., 2016) provides 2,975 training images and 500 validation images of high-quality urban street scenes. **(2) Foggy-Cityscapes** (Sakaridis et al., 2018) extends Cityscapes by simulating fog in the same set of images, maintaining identical annotations to assess robustness under degraded visibility. **(3) BDD100K** (Yu et al., 2020) is a diverse, large-scale dataset with 100,000 driving images; in our experiments, we used the daytime subset containing 36,728 training and 5,258 validation images. **(4) Sim10K** (Johnson-Roberson et al., 2016) offers 10,000 synthetic images, providing a large-scale synthetic-to-real adaptation benchmark. **(5) KITTI** (Geiger et al., 2013) includes 7,481 real-world driving images captured from an onboard camera in urban environments.

## 4.2 IMPLEMENTATION DETAILS

For source-domain training, we employ the official pretrained weights and follow the default optimizer and learning rate settings (Zhao et al., 2024). The **HSA** module was pretrained on the COCO dataset and further updated jointly with the network, with $\lambda_{rec} = 1$.

For target-domain adaptation, we adopt the *teacher-student* paradigm with both the **HSA** and the **CGSC** module, with $\lambda_{rec} = 1$ and $\lambda_{con} = 0.05$. During evaluation, we report performance using the mAP@50 metric. We use RT-DETR (Zhao et al., 2024) as our base detector. All experiments in both the source pretraining and target adaptation stages are conducted on 4 NVIDIA A100 GPUs. More details are provided in the Appendix B.

Table 2: Adaptation from Normal to Foggy Weather: Cityscapes → Foggy-Cityscapes.

| Method | SF | Base | Person | Rider | Car | Truck | Bus | Train | Mcycle | Bicycle | mAP |
|---|---|---|---|---|---|---|---|---|---|---|---|
| Oracle | - | DETR | 55.6 | 55.8 | 75.2 | 44.8 | 63.5 | 56.1 | 45.6 | 52.2 | 56.1 |
| DA-Faster(Chen et al., 2018) | ✗ | FRCNN | 29.2 | 40.4 | 43.4 | 19.7 | 38.3 | 28.5 | 23.7 | 32.7 | 32.0 |
| SFA(Wang et al., 2021) | ✗ | DETR | 46.5 | 48.6 | 62.6 | 25.1 | 46.2 | 29.4 | 28.3 | 44.0 | 41.3 |
| AQT(Huang et al., 2022) | ✗ | DETR | 49.3 | 52.3 | 64.4 | 27.7 | 53.7 | 46.5 | 36.0 | 46.4 | 47.1 |
| AT(Li et al., 2022b) | ✗ | FRCNN | 43.7 | 54.1 | 62.3 | 31.9 | 54.4 | 49.3 | 35.2 | 47.9 | 47.4 |
| MRT(Zhao et al., 2023) | ✗ | DETR | 52.8 | 51.7 | 68.7 | 35.9 | 58.1 | 54.5 | 41.0 | 47.1 | 51.2 |
| CAT(Kennerley et al., 2024) | ✗ | FRCNN | 44.6 | 57.1 | 63.7 | 40.8 | 66.0 | 49.7 | 44.9 | 53.0 | 52.5 |
| DATR(Chen et al., 2025) | ✗ | DETR | 61.6 | 60.4 | 74.3 | 35.7 | 60.3 | 35.4 | 43.6 | 55.9 | 53.4 |
| SFOD(Li et al., 2021) | ✓ | FRCNN | 21.7 | 44.0 | 40.4 | 32.6 | 11.8 | 25.3 | 34.5 | 34.3 | 30.6 |
| SFOD-M(Li et al., 2021) | ✓ | FRCNN | 25.5 | 44.5 | 40.7 | 33.2 | 22.2 | 28.4 | 34.1 | 39.0 | 33.5 |
| LODS(Li et al., 2022a) | ✓ | FRCNN | 34.0 | 45.7 | 48.8 | 27.3 | 39.7 | 19.6 | 33.2 | 37.8 | 35.8 |
| A$^2$SFOD(Chu et al., 2023) | ✓ | FRCNN | 32.3 | 44.1 | 44.6 | 28.1 | 34.3 | 29.0 | 31.8 | 38.9 | 35.4 |
| IRG(VS et al., 2023) | ✓ | FRCNN | 37.4 | 45.2 | 51.9 | 24.4 | 39.6 | 25.2 | 31.5 | 41.6 | 37.1 |
| PETS(Liu et al., 2023) | ✓ | FRCNN | 46.1 | 52.8 | 63.4 | 21.8 | 46.7 | 25.5 | 37.4 | 48.4 | 40.3 |
| LPLD(Yoon et al., 2024) | ✓ | FRCNN | 39.7 | 49.1 | 56.6 | 29.6 | 46.3 | 26.4 | 36.1 | 43.6 | 40.9 |
| SF-UT(Hao et al., 2024) | ✓ | FRCNN | 40.9 | 48.0 | 58.9 | 29.6 | 51.9 | 50.2 | 36.2 | 44.1 | 45.0 |
| TITAN(Ashraf & Bashir, 2025) | ✓ | DETR | **52.8** | 51.7 | 68.0 | 43.2 | **65.5** | 41.8 | **46.0** | 48.7 | 52.2 |
| **CGSA(Ours)** | ✓ | DETR | 49.6 | **53.1** | **68.6** | **43.7** | 62.1 | **54.3** | 44.5 | **49.3** | **53.2** |

## 4.3 MAIN RESULTS

**Cityscapes → BDD100K.** We evaluate our method under the challenging domain adaptation scenario from a small-scale, homogeneous dataset (Cityscapes) to a large-scale, diverse dataset (BDD100K), as presented in Table 1. Our approach outperforms the state-of-the-art (SOTA) source-free methods by nearly 15%, and also surpasses leading traditional DAOD methods by around 10%. This significant improvement demonstrates the strong generalization capability of our framework, particularly when transferring from limited to large-scale data. These advantages indicate the potential for future real-world deployment in more complex and scalable domain adaptation tasks.

Table 3: Adaptation from Synthetic to Real Images: Sim10K → Cityscapes, Adaptation on cross-camera: KITTI → Cityscapes. The results are reported using the "car" class.

| Method | SF | Base | S2C | K2C |
|---|---|---|---|---|
| Oracle | - | DETR | 79.9 | 79.9 |
| DA-Faster(Chen et al., 2018) | ✗ | FRCNN | 41.9 | 41.8 |
| SFA(Wang et al., 2021) | ✗ | DETR | 52.6 | 41.3 |
| DATR(Chen et al., 2025) | ✗ | DETR | 64.3 | 53.3 |
| SFOD(Li et al., 2021) | ✓ | FRCNN | 42.3 | 43.6 |
| SFOD-M(Li et al., 2021) | ✓ | FRCNN | 42.9 | 44.6 |
| IRG(VS et al., 2023) | ✓ | FRCNN | 45.2 | 46.9 |
| A$^2$SFOD(Chu et al., 2023) | ✓ | FRCNN | 44.0 | 44.9 |
| PETS(Liu et al., 2023) | ✓ | FRCNN | 57.8 | 47.0 |
| LPLD(Yoon et al., 2024) | ✓ | FRCNN | 49.4 | 51.3 |
| SF-UT(Hao et al., 2024) | ✓ | FRCNN | 55.4 | 46.2 |
| TITAN(Ashraf & Bashir, 2025) | ✓ | DETR | 59.8 | 53.2 |
| **CGSA(Ours)** | ✓ | DETR | **67.7** | **60.8** |

**Cityscapes → Foggy-Cityscapes.** We evaluate our method under the classic adaptation scenario from clear weather (Cityscapes) to foggy conditions (Foggy-Cityscapes), as presented in Table 2. Our approach outperforms all existing SF-DAOD methods and surpasses the majority of traditional DAOD methods. These results demonstrate our modules effectively disentangle object-level features in the target domain and guide them toward category-consistent semantics, leading to more robust adaptation under weather degradation.

**Sim10K → Cityscapes.** This experiment evaluates synthetic-to-real domain adaptation from Sim10K to Cityscapes, focusing on the car category. As shown in Table 3, our method achieves the best performance among all approaches in two settings. The results demonstrate that even in single-class adaptation sce-

narios (see Appendix D for implementation details), our model can achieve state-of-the-art performance.

**KITTI → Cityscapes.** This experiment evaluates from KITTI to Cityscapes, focusing on the car category across real-world datasets with different viewpoints and camera setups. As shown in Table 3, our method also achieves the best performance among all approaches in two settings. For the single-class setting, the implementation details of Section 3.3 are provided in the Appendix D.

## 4.4 ABLATION STUDY

### 4.4.1 ABLATION ON COMPONENTS.

To evaluate the contribution of each component in our framework, we conduct an ablation study over all datasets, as shown in Figure 4. The results show that incorporating **HSA** yields consistent improvements across all datasets, demonstrating that structural, slot-based visual priors play a crucial role in stabilizing target-domain localization. Building on this, **CGSC** consistently brings additional gains by aligning slot representations with class semantics in a domain-invariant manner. The improvements are stable across diverse domain shifts, which suggests that both components are complementary, producing stronger source-free adaptation overall. See the Appendix F& G for additional ablation experiments.

### 4.4.2 ABLATION ON THE HYPERPARAMETER OF HSA.

As shown in Table 4, when we use two-level HSA (depth=2), the performance is not highly sensitive to the slot number. Across a wide range of $number \in \{2, 4, 5, 6, 8, 10\}$, mAP stays within a relatively small and acceptable band, showing robustness to this hyperparameter. The observed trend is both theoretically and intuitively justified, exhibiting a pattern that is "low at both ends, high in the middle", where very small number (too few slots, e.g., 4 total slots) cannot capture sufficient object structure, while very large numbers(too many slots, e.g., 64 and 100 total slots) dilutes object-centric grouping and tends to fragment objects, thus reducing the usefulness of the structural prior.

Table 4: Hyperparameter Sensitivity Analysis. "Depth" denotes the hierarchy depth of HSA, and "Number" is the slot count per level. The total number of fine slots is $Number^{Depth}$.

| Depth | Number | Total Slot Numbers | mAP |
|-------|--------|--------------------|------|
| 2 | 2 | 4 | 50.9 |
| 2 | 4 | 16 | 52.9 |
| 2 | 5 | 25 | **53.2** |
| 2 | 6 | 36 | 52.6 |
| 2 | 8 | 64 | 52.5 |
| 2 | 10 | 100 | 51.2 |
| 3 | 2 | 8 | 50.8 |
| 3 | 3 | 27 | 50.3 |
| 3 | 4 | 64 | 48.7 |

Increasing the hierarchy depth to 3 leads to a clear performance drop. We attribute this to the unsupervised nature of slot attention, since HSA is trained only with self-supervision, and deeper hierarchies weaken the reconstruction constraint at each level and accumulate decomposition errors, resulting in lower-quality slots for adaptation. Finally, when Depth = 1, HSA degenerates to standard Slot Attention. As discussed in Appendix G, training becomes unstable and often collapses when the number is large, which motivates our hierarchical design. Therefore, considering both robustness and performance, we choose Depth = 2 and Number=5 as our default setting, which gives the highest mAP of 53.2.

## 4.5 ANALYSIS

### 4.5.1 VISUALIZATION OF QUERY

Figure 5 compares the target-domain feature distributions of the "Source Only" baseline and our approach. Unlike the scattered and overlapping features in the baseline, our approach produces clearer inter-class separation and tighter intra-class clustering. This suggests that **HSA** provides structured visual priors that stabilize feature formation, while **CGSC** pulls slot representations toward class prototypes, improving semantic alignment in the target domain. The resulting geometry is consistent with higher mAP and supports the effectiveness of the proposed components.

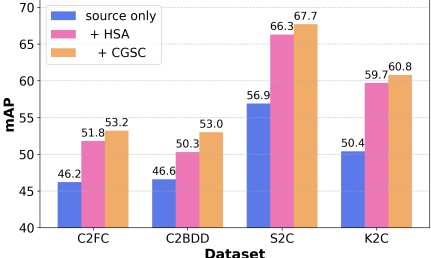 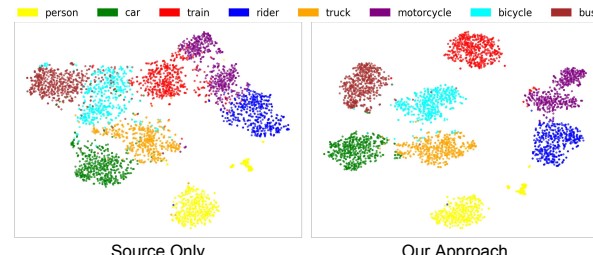

Figure 4: Ablation on four cross-domain benchmarks. "Source Only" denotes the baseline without adaptation, where the source-trained model is directly tested on the target domain.

Figure 5: The t-SNE visualization comparing feature distributions of object queries on the Foggy-Cityscapes dataset between "Source Only" and our approach.

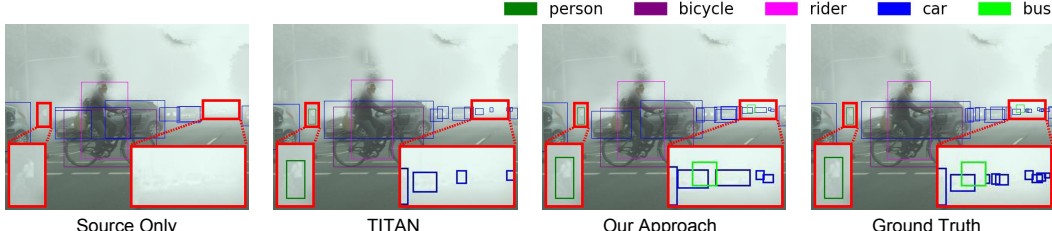

Figure 6: Qualitative comparison of our method with the "Source Only" and TITAN. The red box is enlarged to highlight and compare the detection results more clearly. Our method captures fine-grained details more effectively, detecting distant objects that the other methods miss.

### 4.5.2 QUALITATIVE RESULTS

Figure 6 provides a visual comparison of the results on the Foggy-Cityscapes. The "Source Only" model detects just the nearby, larger objects. Although TITAN identifies the person on the left, it misses some distant objects on the right. In contrast, our approach **CGSA** performs better, successfully detecting the distant cars and bus obscured by dense fog and yielding more accurate bounding boxes for the objects in the foreground. Notably, since the official TITAN implementation is not publicly available, we re-implemented the method and report results from our reproduction.

## 5 LIMITATIONS

Our approach is tailored to DETR-based detectors because OCL and DETR share the same attention-based computation, making their underlying logic consistent. Portability to two-stage or non-query architectures remains to be established. In the transfer setting, we explore only object detection. More vision tasks, such as classification and segmentation, should be covered in future work.

## 6 CONCLUSION

We introduce **CGSA**, the first SF-DAOD framework that integrates Object-Centric Learning into a DETR-based detector by decomposing target images into hierarchically refined slot-aware priors, and guiding them with semantics via class-guided contrast. Extensive experiments and in-depth analysis demonstrate the effectiveness and robustness of the proposed components and the generality of the framework. The object-centric paradigm also opens a promising path for the privacy-preserving domain transfer.

ACKNOWLEDGMENTS

This work is supported in part by Hong Kong Research Grants Council under NSFC/RGC Collaborative Research Scheme (Grant CRS HKU703/24) and the focused project on spatial intelligence at Shenzhen Loop Area Institute.

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

APPENDIX

# A PROOF

*Why Guided Slots Form Domain-Invariant Priors and Yield Convergent Improvements in Source-Free Target Adaptation?*

## A.1 STANDING SETTING AND NOTATION.

We consider a $C$-class detection task with image space $\mathcal{X} \subset \mathbb{R}^{H \times W \times 3}$ and label space $\mathcal{Y}$ of finite sets of pairs $(b, c)$ with $b \in [0,1]^4$ and $c \in \{1, \ldots, C\}$. The source and target joint distributions factorize as $P_{XY}^S = P_X^S P_{Y|X}^S$ and $P_{XY}^T = P_X^T P_{Y|X}^T$.

The learning protocol contains two phases. On the source pretraining, a DETR-based detector $f_\theta$ is trained on $\mathcal{D}_S$ by the standard detection loss, yielding $\theta_0$. On the target adaptation, only $\mathcal{D}_T \sim P_X^T$ is available; we maintain a student–teacher pair $(\theta_s, \theta_t)$ initialized by $\theta_0$ and update the teacher by EMA $\theta_t \leftarrow \gamma \theta_t + (1 - \gamma) \theta_s$. Teacher predictions are turned into training targets via $G(\widehat{\mathcal{Y}}(x); \kappa)$, and the student minimizes $\mathbb{E}_{x \sim P_X^T}[\ell_{\det}(f_{\theta_s}(x), G(\widehat{\mathcal{Y}}(x); \kappa))]$.

We follow all the modules, losses, and notation as defined in the main text: hierarchical slot construction and reconstruction losses (Eqs. 1–3), slot-aware query fusion (Eq. 4), class prototype maintenance (Eqs. 5–6), weighted slots, matching, and contrastive training (Eqs. 7–9), and the target-domain objective (Eqs. 10–11). The hypotheses H1–H3 below are assumed in all proofs.

## A.2 HYPOTHESES

- **H1 Bounded background**: the backbone feature can be decomposed as $h = h^{\mathrm{obj}} + h^{\mathrm{bg}}$ with $\mathbb{E}[h^{\mathrm{bg}} \mid X] = 0$ and $\mathrm{Cov}(h^{\mathrm{bg}}) \preceq \sigma^2 I$.
- **H2 Reconstruction consistency**: the hierarchical reconstruction objective in Eq. 3 induces a slot projection $S(h)$ that is approximately identity on the object subspace $\mathcal{S}$, i.e., $\mathbb{E}\|S(h) - \Pi_{\mathcal{S}}(h^{\mathrm{obj}})\|^2 \leq \epsilon_{\mathrm{rec}}$.
- **H3 Prototype semantic stability**: the class anchors $P_c$ maintained by Eqs. 5 & 6 have bounded drift across iterations, and enjoy a positive cosine margin w.r.t. same-class weighted slots.

## A.3 LEMMA 1 (VARIANCE CONTRACTION: WEIGHTED SLOTS SUPPRESS DOMAIN-SPECIFIC BACKGROUND)

**Claim.** Let the weighted slot $\tilde{z}_k$ be defined as in Eq. 7. Under H1 & H2,

$$\mathbb{E}\left\|\tilde{z}_k - h_{\mathrm{eff}}^{\mathrm{obj}}\right\|^2 \;\leq\; \epsilon_{\mathrm{rec}} \;+\; \kappa_k \sigma^2, \qquad \kappa_k \;:=\; \left\|m_k^{(2)}\right\|_2^2 \in (0,1]. \tag{A.1}$$

Here $h_{\mathrm{eff}}^{\mathrm{obj}}$ denotes the effective component of $h$ in the object subspace $\mathcal{S}$. The quantity $\kappa_k = \|m_k^{(2)}\|_2^2$ acts as a clear contraction factor.

*Proof.* Write $h = h^{\mathrm{obj}} + h^{\mathrm{bg}}$ as in H1. Then by the convex-variance bound,

$$\mathrm{Var}\left((m_k^{(2)})^\top h^{\mathrm{bg}}\right) \;\leq\; \left\|m_k^{(2)}\right\|_2^2 \sigma^2 \;=\; \kappa_k \sigma^2. \tag{A.2}$$

By the reconstruction consistency in H2, the slot projection $S(h)$ is approximately identity on $\mathcal{S}$, i.e., $\mathbb{E}\|S(h) - \Pi_{\mathcal{S}}(h^{\mathrm{obj}})\|^2 \leq \epsilon_{\mathrm{rec}}$. Identifying $S(h)$ with the weighted aggregation above yields equation A.1. This completes the proof.

## A.4 LEMMA 2 (INFONCE ENLARGES COSINE INTER-CLASS MARGINS AND SUPPRESSES CROSS-CLASS SIMILARITY)

**Claim.** Let the contrastive loss between class prototypes and slot prototypes be Eq. 9, i.e., an InfoNCE/NT-Xent objective with positive pairs $(P_c, \bar{z}_c)$ and negatives $(P_c, \bar{z}_{c'})$. Then the gradients w.r.t. the similarities satisfy

$$\frac{\partial \mathcal{L}_{\mathrm{con}}}{\partial s_{\mathrm{pos}}} = -\frac{1 - \pi_{\mathrm{pos}}}{\tau} < 0, \qquad \frac{\partial \mathcal{L}_{\mathrm{con}}}{\partial s_{\mathrm{neg}}} = \frac{\pi_{\mathrm{neg}}}{\tau} > 0, \tag{A.3}$$

where $\pi$ are softmax weights and $\tau$ is the temperature. Consequently, a gradient-descent step on $\mathcal{L}_{\text{con}}$ *increases* the same-class similarity and *decreases* the cross-class similarities, thereby enlarging the cosine margin

$$\Delta_t := \Big( \mathbb{E}\,\varphi(P_c, \tilde{z} \mid Y = c) - \max_{c' \neq c} \mathbb{E}\,\varphi(P_{c'}, \tilde{z} \mid Y = c) \Big). \tag{A.4}$$

*Proof.* Write the InfoNCE loss in Eq. 9 for one anchor $P_c$ as

$$\begin{aligned}
\mathcal{L}_{\text{con}} &= -\log\left( \frac{\exp(s_{\text{pos}}/\tau)}{\exp(s_{\text{pos}}/\tau) + \sum_{c' \neq c} \exp(s_{c'}^{\text{neg}}/\tau)} \right) \\
&= -\frac{s_{\text{pos}}}{\tau} + \log\Big( \exp(s_{\text{pos}}/\tau) + \sum_{c' \neq c} \exp(s_{c'}^{\text{neg}}/\tau) \Big),
\end{aligned} \tag{A.5}$$

where $s_{\text{pos}} := \varphi(P_c, \bar{z}_c)$ and $s_{c'}^{\text{neg}} := \varphi(P_c, \bar{z}_{c'})$. Let

$$Z := \exp(s_{\text{pos}}/\tau) + \sum_{c' \neq c} \exp(s_{c'}^{\text{neg}}/\tau), \qquad \pi_{\text{pos}} := \frac{\exp(s_{\text{pos}}/\tau)}{Z}, \quad \pi_{c'} := \frac{\exp(s_{c'}^{\text{neg}}/\tau)}{Z}. \tag{A.6}$$

Then

$$\frac{\partial \mathcal{L}_{\text{con}}}{\partial s_{\text{pos}}} = -\frac{1}{\tau} + \frac{1}{Z} \cdot \frac{1}{\tau} \exp(s_{\text{pos}}/\tau) = -\frac{1 - \pi_{\text{pos}}}{\tau} < 0, \tag{A.7}$$

and for any negative $c' \neq c$,

$$\frac{\partial \mathcal{L}_{\text{con}}}{\partial s_{c'}^{\text{neg}}} = \frac{1}{Z} \cdot \frac{1}{\tau} \exp(s_{c'}^{\text{neg}}/\tau) = \frac{\pi_{c'}}{\tau} > 0. \tag{A.8}$$

Therefore, under a gradient-descent update $s \leftarrow s - \eta\, \partial L_{\text{con}}/\partial s$ with step size $\eta > 0$, we have $s_{\text{pos}}$ strictly increases and each $s_{c'}^{\text{neg}}$ strictly decreases. Since the cosine margin $\Delta_t$ is the difference between the expected same-class similarity and the maximal cross-class similarity, these monotone updates enlarge $\Delta_t$ (in expectation over samples). The construction of $(P_c, \bar{z}_c)$ via Eqs. 5- 8 ensures semantic stability of positives and well-defined negatives across steps. The gradient form above is the standard derivative of the log-softmax used in CPC/SimCLR analyses (Oord et al., 2018; Chen et al., 2020). This completes the proof.

## A.5 LEMMA 3 (MARGIN TRANSFER: SLOT-AWARE QUERY INFUSION YIELDS AT LEAST LINEAR DECODER-MARGIN GAIN)

**Claim.** Let the decoder $g$ be locally $L$-Lipschitz in its query input, and let $Q_{\text{aware}} = Q_{\text{obj}} + \tilde{Z}$ as in Eq. 4. We use the shorthand $\tilde{Z} := \tilde{z}^{(2)}$ for the stack of mapped level-2 slots. Assume moreover that on the subspace $\mathcal{S} := \text{span}\{P_c\}$ the map $Q \mapsto g(Q)$ is locally *bi-Lipschitz* with constants $(\mu, L)$. Then

$$\big| \text{margin}_{\text{dec}}(Q_{\text{aware}}) - \text{margin}_{\text{dec}}(Q_{\text{obj}}) \big| \geq \mu\, \|\tilde{Z}\|_{\text{eff}}, \tag{A.9}$$

where $\|\tilde{Z}\|_{\text{eff}}$ denotes the norm of the component of $\tilde{Z}$ aligned with the span of $\{P_c\}$.

*Proof.* Let $\phi(Q) := \text{margin}_{\text{dec}}(Q)$ denote the decoder-level margin, i.e., $\phi(Q) = s_{\text{pos}}(Q) - \max_{c' \neq c} s_{c'}^{\text{neg}}(Q)$, where each similarity score $s(Q) = \varphi(P, g(Q))$ is composed with $g$. Write $\Pi$ for the orthogonal projector onto the prototype-aligned subspace $\mathcal{S} := \text{span}\{P_c\}$ and decompose $\tilde{Z} = \tilde{Z}_{\text{eff}} + \tilde{Z}_\perp$ with $\tilde{Z}_{\text{eff}} := \Pi\tilde{Z}$. Since $g$ is locally $L$-Lipschitz, every score $s(Q)$ is locally $L$-Lipschitz in $Q$, and so is $\phi(Q)$ up to the 1-Lipschitz max operator. Assume moreover that on the subspace $\mathcal{S}$ the map $Q \mapsto g(Q)$ is locally *bi-Lipschitz* with constants $(\mu, L)$, i.e., for all small $\Delta \in \mathcal{S}$, $\mu\|\Delta\| \leq \|g(Q + \Delta) - g(Q)\| \leq L\|\Delta\|$. (When the Jacobian $J_g(Q)$ restricted to $\mathcal{S}$ is nonsingular, such $\mu > 0$ exists by the inverse function theorem. See, e.g., (Rudin, 1976).)

By the mean value theorem for vector-valued functions (Rudin, 1976),

$$\phi(Q_{\text{obj}} + \tilde{Z}) - \phi(Q_{\text{obj}}) = \int_0^1 \langle \nabla\phi(Q_{\text{obj}} + t\tilde{Z}),\, \tilde{Z} \rangle\, dt. \tag{A.10}$$

For a lower bound, it suffices to consider the projection of $\tilde{Z}$ onto $\mathcal{S}$; hence we work with $\tilde{Z}_{\mathrm{eff}} := \Pi \tilde{Z}$ and drop the orthogonal component:

$$\left| \phi(Q_{\mathrm{obj}} + \tilde{Z}) - \phi(Q_{\mathrm{obj}}) \right| \ \geq \ \left| \int_0^1 \langle \nabla \phi(Q_{\mathrm{obj}} + t\tilde{Z}), \tilde{Z}_{\mathrm{eff}} \rangle \, dt \right| \ \geq \ \inf_{t \in [0,1]} \|\nabla \phi(Q_{\mathrm{obj}} + t\tilde{Z})\| \, \|\tilde{Z}_{\mathrm{eff}}\|. \tag{A.11}$$

Along $\mathcal{S}$, variations of $\phi$ are controlled from below by the bi-Lipschitz constant of $g$; in particular, we obtain

$$\left| \phi(Q_{\mathrm{obj}} + \tilde{Z}) - \phi(Q_{\mathrm{obj}}) \right| \ \geq \ \mu \, \|\tilde{Z}_{\mathrm{eff}}\|. \tag{A.12}$$

Recalling that $\|\tilde{Z}\|_{\mathrm{eff}} := \|\tilde{Z}_{\mathrm{eff}}\|$ yields the claim. This completes the proof.

## A.6 THEOREM (A BASIC RISK DESCENT INEQUALITY ON THE TARGET DOMAIN)

Let the target risk $\mathcal{R}_T(\theta)$ be the expectation of the target-domain detection objective in Eq. 10 plus regularization in Eq. 11. Under H1-H3 and per-step minimization, there exist constants $c_1, c_2 > 0$ such that

$$\mathbb{E} \, \mathcal{R}_T(\theta_{t+1}) \ \leq \ \mathbb{E} \, \mathcal{R}_T(\theta_t) \ - \ c_1 \, \Delta_t \ + \ c_2 \, (\epsilon_{\mathrm{rec}} + \sigma^2). \tag{A.13}$$

*Proof.* We decompose the one-step change of the expected risk into (i) a margin-driven decrease in the classification component and (ii) a residual term due to representation noise.

(i) *Margin gain.* By Lemma 2, the InfoNCE objective in Eq. 9 yields monotone updates that increase the same-class similarity and decrease cross-class similarities, enlarging the cosine margin $\Delta_t$ on the slot–prototype pairs. By Lemma 3, this margin gain transfers through the decoder at least linearly. Hence there exists $a_1 > 0$ such that the classification part of the risk satisfies

$$\mathbb{E} \, \mathcal{L}_T^{\mathrm{cls}}(\theta_{t+1}) \ \leq \ \mathbb{E} \, \mathcal{L}_T^{\mathrm{cls}}(\theta_t) \ - \ a_1 \, \Delta_t. \tag{A.14}$$

(ii) *Residual noise.* By Lemma 1, the weighted-slot aggregation contracts the background variance, and the reconstruction consistency H2 bounds the object-subspace approximation error by $\epsilon_{\mathrm{rec}}$. Consequently, the representation noise contributes at most a term $a_2(\epsilon_{\mathrm{rec}} + \sigma^2)$ to the change of the total risk, for some $a_2 > 0$ that absorbs Lipschitz constants of the detection loss and the decoder.

Combining (i) and (ii), and noting that Eqs. 10 & 11 just add Lipschitz-continuous regularization terms, we arrive at

$$\mathbb{E} \, \mathcal{R}_T(\theta_{t+1}) \ \leq \ \mathbb{E} \, \mathcal{R}_T(\theta_t) \ - \ a_1 \, \Delta_t \ + \ a_2 \, (\epsilon_{\mathrm{rec}} + \sigma^2). \tag{A.15}$$

Renaming $c_1 := a_1$ and $c_2 := a_2$ gives equation A.13. This completes the proof.

## A.7 COROLLARY (CONVERGENCE VIA CONTRACTION)

**Notation.** Let $\eta_t \in [0,1]$ denote an effective noise level (e.g., an upper-bound proxy of classification/matching error on the target domain) at step $t$, and let $\Delta_t \geq 0$ be the margin increment defined above. Consider the iteration

$$\Phi : \ \eta_t \mapsto \eta_{t+1} \ = \ \eta_t \ - \ \alpha \, \Delta_t \ + \ \beta \, (\epsilon_{\mathrm{rec}} + \sigma^2), \tag{A.16}$$

where $\alpha > 0$ summarizes the efficiency of contrastive/prototype alignment and query infusion, and $\beta \geq 0$ captures residual effects of reconstruction error and background variance. By Lemmas 2–3, there exists $k > 0$ such that $\Delta_t \geq k \, (\eta_t - \eta^\star)_+$ for some fixed point $\eta^\star$.

**Claim.** If $0 < \alpha k < 1$ and $\beta(\epsilon_{\mathrm{rec}} + \sigma^2) \leq (1 - \alpha k) \, \eta^\star$, then $\Phi$ is a contraction on $[0,1]$:

$$|\Phi(\eta) - \Phi(\eta')| \ \leq \ (1 - \alpha k) \, |\eta - \eta'| \qquad (\forall \, \eta, \eta' \in [0,1]). \tag{A.17}$$

Hence, by Banach's fixed-point theorem (Banach, 1922), $\eta_t$ converges to the unique fixed point $\eta^\star$. Together with the risk inequality equation A.13 and $\Delta_t \geq k(\eta_t - \eta^\star)_+$, $\mathbb{E} \, \mathcal{R}_T(\theta_t)$ decreases monotonically and converges.

*Proof.* By definition, $|\Phi(\eta) - \Phi(\eta')| = \alpha \, |\Delta(\eta) - \Delta(\eta')| \leq \alpha k \, |\eta - \eta'|$. Let $\rho := \alpha k < 1$; contraction follows. Banach's fixed-point theorem then yields existence/uniqueness and geometric convergence. Summing equation A.13 over $t$ yields risk convergence. This completes the proof.

# B EXPERIMENTAL DETAILS

## B.1 SOURCE-DOMAIN PRETRAINING

During the source pretraining stage, we train the detector only on source-domain images. The training is based on RT-DETR (Zhao et al., 2024) with official pretrained weights on COCO dataset and its default optimizer and learning rate configurations. The **HSA** module, pretrained for 100 epochs on the COCO dataset, is included in the training process, and the reconstruction loss is applied with a weight of $\lambda_{rec} = 1$. Training is conducted for 72 epochs with gradient clipping at a maximum norm of 0.1. Mixed precision and exponential moving average are enabled, where EMA is configured with a decay of 0.9999 and a warmup of 2000 steps.

The optimizer is AdamW. Backbone layers excluding normalization are trained with a learning rate of 1e-5, encoder and decoder normalization layers are set with zero weight decay, and the remaining parameters use a base learning rate of 1e-4 with $\beta = (0.9, 0.999)$ and weight decay of 1e-4. The **HSA** module is optimized with a separate learning rate of 1e-5. Learning rate scheduling follows a MultiStepLR strategy with milestones at 1000 iterations and a decay factor of 0.1, together with a linear warmup scheduler of 2000 steps.

## B.2 TARGET-DOMAIN ADAPTATION

During the target adaptation stage, we adopt a *teacher-student* architecture and run 12 epochs. The teacher is updated by an exponential moving average with decay 0.9993, initialized from the source-pretrained student.

The training objective in the **CGSA** component adds a contrastive term with coefficient $\lambda_{con} = 0.05$. All other settings are identical to the source stage, including the use of **HSA** (with its reconstruction loss weight $\lambda_{rec} = 1$ and its learning rate of $1 \times 10^{-5}$), the RT-DETR defaults, optimizer and parameter groups (AdamW with the same learning rates, betas, and weight decay), learning-rate scheduling and warmup, mixed precision, gradient clipping, and any remaining implementation details. All experiments in both the source pretraining and target adaptation stages are conducted on 4 NVIDIA A100 GPUs.

During inference, the **HSA** remains active while **CGSC** is turned off, so the parameter count increases only marginally and the model does not become overly large.

# C DYNAMIC THRESHOLD SELECTION

During target-domain adaptation, following previous work, we employ a standard *teacher-student* framework. At each step, the *teacher* network generates pseudo labels $\hat{y}$ for a mini-batch of target images $\mathcal{X}_t$; the *student* network is trained with these labels and updates its parameters $\theta_s$ via backpropagation, while the teacher parameters $\theta_t$ are updated with an exponential moving average (EMA):

$$\theta_t \leftarrow \gamma\,\theta_t + (1-\gamma)\,\theta_s, \gamma \in [0,1). \tag{C.1}$$

Because supervision entirely relies on pseudo labels, selecting reliable ones is critical to stable adaptation. Rather than fixing a single confidence threshold, we propose a cosine-adaptive strategy that gradually relaxes the threshold as the student improves. Denote the current optimization step by $s$ and the total number of adaptation steps by $S$. The threshold $\tau(s) \in [\tau_{\min}, \tau_{\max}]$ is computed as:

$$\tau(s) = \tau_{\min} + (\tau_{\max} - \tau_{\min}) \cdot \frac{1 + \cos(\pi\,s/S)}{2}. \tag{C.2}$$

This schedule starts with a strict filter ($\tau \approx \tau_{\max}$) to suppress early noise, then gradually lowers the bar, allowing more pseudo labels to participate once the detector becomes more accurate. The smooth, non-monotonic decay avoids abrupt changes that could destabilize training and requires no hand-tuned milestones. We also construct parameter ablation and comparative experiments, see Appendix F.

# D  SINGLE-CLASS PROTOCOL ON CLASS-GUIDED SLOT CONTRAST

In the datasets SIM10K→CITYSCAPES and KITTI→CITYSCAPES, detection is conducted under a *single-class* protocol focusing exclusively on the *car* category. Concretely, SIM10K provides bounding boxes only for cars, while CITYSCAPES contains multiple traffic-related categories but evaluation is restricted to cars. Similarly, when transferring from KITTI to CITYSCAPES, we retain only the car annotations and treat all non-car instances as background. This setting isolates domain shift without confounding class imbalance or multi-class interference, and it aligns with prevalent practice in the DA detection literature.

Under this single-class setting, our slot–query matching and prototype construction simplify accordingly. Let $\tilde{z}_k$ be the $k$-th weighted slot and $q_i$ the $i$-th decoder query. We compute cosine similarities

$$S_{ki} = \phi(\tilde{z}_k, q_i), \tag{D.1}$$

and obtain a one-to-one assignment $\pi(k)$ via the Hungarian algorithm applied to $S$. Since every valid detection hypothesis is of the same semantic class (car), the assignment induces a trivial label for each slot, i.e.,

$$\hat{c}_k = \texttt{car} \quad \text{for all } k, \tag{D.2}$$

so that all slots fall into a single index set

$$\mathcal{Z}_{\texttt{car}} = \big\{\, k \mid \hat{c}_k = \texttt{car} \,\big\} = \{1, 2, \ldots, K\}. \tag{D.3}$$

We then form the single-class slot prototype by averaging all weighted slots:

$$\bar{z}_{\texttt{car}} = \frac{1}{|\mathcal{Z}_{\texttt{car}}|} \sum_{k \in \mathcal{Z}_{\texttt{car}}} \tilde{z}_k. \tag{D.4}$$

The contrastive regulation in the multi-class case, where each class-specific slot prototype is pulled toward its corresponding global prototype and pushed away from the remaining prototypes, reduces here to a pure alignment objective, because the InfoNCE denominator collapses when $C = 1$. Let $P_{\texttt{car}}$ denote the global prototype of the car class. The single-class contrastive loss becomes

$$\mathcal{L}_{\text{con}}^{\text{single}} = -\phi\big(P_{\texttt{car}}, \bar{z}_{\texttt{car}}\big), \tag{D.5}$$

which maximizes the cosine agreement between the dataset-level car prototype and the slot-level car prototype. Intuitively, this objective preserves the attractive force that consolidates car-specific semantics across domains while discarding inter-class repulsion that is meaningless in a single-class regime, thereby stabilizing representation learning and ensuring that the slots encode domain-invariant, class-relevant information for cars. In implementation, we set $C=1$ so that the label indices become degenerate. We compute a single dataset prototype of dimension $D$, and replace the multi-class InfoNCE with the alignment objective above. All other components, including slot weighting, similarity-based matching, and exponential moving average updates of the dataset prototype, remain unchanged and operate identically under the single-class constraint.

# E  PSEUDOCODE

Below is the pseudocode from Section 3.2.1 & 3.2.2 for reference.

---

**Algorithm 1** Hierarchical Slot-Based Reconstruction

---

**Require:** Input features $h$, token length $N$, per token feature dimension $d$, number of slots $n$, slot dimension $d$
1: **Stage 1: Coarse Decomposition** ▷ obtain coarse slots
2: $z^{(1)} \leftarrow$ Encode $h$ with slot attention
3: Broadcast $z^{(1)}$ across tokens
4: $(\hat{h}^{(1)}, m^{(1)}) \leftarrow$ Decode into reconstruction and masks
5: Split $\hat{h}^{(1)}$ into $n$ slot-specific feature maps
6: **Stage 2: Fine Decomposition** ▷ refine into fine slots
7: Initialize $z^{(2)} \leftarrow$ Gaussian distribution
8: **for** each coarse slot feature map **do**
9:     Get fine slots $\leftarrow$ Encode with slot attention
10:     Append to $z^{(2)}$
11: **end for**
12: Concatenate into $n^2$ slots
13: $(\hat{h}^{(2)}, m^{(2)}) \leftarrow$ Broadcast $z^{(2)}$ and decode
14: **Stage 3: Weighted Slot Features** ▷ aggregate features
15: Normalize masks $m^{(2)}$ across tokens
16: $w^{(2)} \leftarrow$ Weighted combination of $m^{(2)}$ and input features $h$
17: **Stage 4: Reconstruction Loss**
18: $\mathcal{L}_{\text{rec}} = \left\| \hat{h}^{(1)} - h \right\|_2^2 + \left\| \hat{h}^{(2)} - h \right\|_2^2.$
19: **Outputs:** $\hat{h}^{(2)}, z^{(2)}, m^{(2)}, w^{(2)}$

---

Below is the pseudocode from Section 3.2.3 for reference.

---

**Algorithm 2** Slot-Aware Target Fusion with MLP

---

**Require:** Object queries $tgt \in \mathbb{R}^{B \times L \times d}$, slots $z \in \mathbb{R}^{B \times n \times d}$, slot mapper $f_{\text{map}} : \mathbb{R}^{B \times 1 \times d} \mapsto \mathbb{R}^{B \times s \times d}$
1: Compute segment length $s \leftarrow L/n$
2: Initialize fused segments $\mathcal{F} \leftarrow \varnothing$
3: **for** $i = 1$ to $n$ **do**
4:     Extract target segment $tgt_i \leftarrow tgt[:, (i-1)s : is, :]$ ▷ $B \times s \times d$
5:     Select slot $z_i \leftarrow z[:, i : i+1, :]$ ▷ $B \times 1 \times d$
6:     Map slot to segment $mapped\_z_i \leftarrow f_{\text{map}}(z_i)$ ▷ $B \times s \times d$
7:     Fuse with target segment $\hat{tgt}_i \leftarrow tgt_i + mapped\_z_i$
8:     Append $\hat{tgt}_i$ to $\mathcal{F}$
9: **end for**
10: Concatenate fused segments $\tilde{tgt} \leftarrow \text{Concat}(\mathcal{F}, \dim = 1)$
11: **return** $\tilde{tgt}$

---

Below is the pseudocode from Section 3.3 for reference.

---

**Algorithm 3** Class-Guided Slot Contrast (CGSC)

---

**Require:** Decoder queries $\{q_i\}_{i=1}^M$ with predicted labels $\hat{c}_i$, weighted slots $\{\tilde{z}_k\}_{k=1}^n$, class proto-
    types $\{P_c\}_{c=1}^C$
 1: **Stage 1: Class-prototype memory**
 2: **for** each class $c = 1 \ldots C$ **do**
 3:     Collect queries $\mathcal{Q}_c \leftarrow \{q_i \mid \hat{c}_i = c\}$
 4:     **if** $\mathcal{Q}_c$ not empty **then**
 5:        Compute temporary prototype $\bar{p}_c \leftarrow \text{mean}(\mathcal{Q}_c)$
 6:        Update global prototype $P_c \leftarrow \beta P_c + (1 - \beta)\bar{p}_c$
 7:     **end if**
 8: **end for**
 9: **Stage 2: Weighted slot construction**
10: **for** each slot $k = 1 \ldots n$ **do**
11:     Aggregate features $\tilde{z}_k \leftarrow \sum_i m_{k,i}^{(2)} \cdot h_i$
12: **end for**
13: **Stage 3: Prototype-slot matching**
14: Compute similarity matrix $S_{ki} \leftarrow \phi(\tilde{z}_k, q_i)$
15: Get one-to-one assignment $\pi(k) \leftarrow$ Apply Hungarian algorithm
16: Assign pseudo label $\hat{c}_k \leftarrow \hat{c}_{\pi(k)}$
17: **Stage 4: Slot prototypes**
18: **for** each class $c = 1 \ldots C$ **do**
19:     Collect slots $\mathcal{Z}_c \leftarrow \{k \mid \hat{c}_k = c\}$
20:     **if** $\mathcal{Z}_c$ not empty **then**
21:        Compute slot prototype $\bar{z}_c \leftarrow \text{mean}(\{\tilde{z}_k\}_{k \in \mathcal{Z}_c})$
22:     **end if**
23: **end for**
24: **Stage 5: Contrastive objective**
25: Compute loss:
$$\mathcal{L}_{\text{con}} = -\frac{1}{C} \sum_{c=1}^C \log \frac{\exp\big(\phi(P_c, \bar{z}_c)/\tau\big)}{\sum_{c'=1}^C \exp\big(\phi(P_c, \bar{z}_{c'})/\tau\big)}$$

26: **return** $\mathcal{L}_{\text{con}}$

---

## F   Ablation: Dynamic Threshold Schedules

We investigate the effect of threshold scheduling on target-domain adaptation under a *teacher-student* self-training framework. After an initial *burn-in* stage, different scheduling functions $\tau(s)$ are used, where $s \in [0, S]$ is the step index after *burn-in* and $S$ is the total number of adaptation steps.

**Fixed threshold**
$$\tau_{\text{fixed}}(s) = \tau_{\text{fix}}. \tag{F.1}$$

**Cosine (our method, see the Appendix C)**

$$\tau_{\cos}(s) = \tau_{\min} + (\tau_{\max} - \tau_{\min}) \cdot \frac{1 + \cos\big(\pi\, s/S\big)}{2}. \tag{F.2}$$

This schedule starts strict ($\tau \approx \tau_{\max}$) and gradually decays to $\tau_{\min}$ in a smooth and symmetric manner.

**Exponential**
$$\tau_{\exp}(s) = \tau_{\min} + (\tau_{\max} - \tau_{\min}) \cdot \exp(-\beta\, s), \tag{F.3}$$
where $\beta > 0$ controls the decay speed.

**Sigmoid**
$$\tau_{\text{sigmoid}}(s) = \tau_{\min} + (\tau_{\max} - \tau_{\min}) \cdot \frac{1}{1 + \exp\big(-k(\frac{s}{S} - \frac{1}{2})\big)}, \tag{F.4}$$
where $k$ controls the steepness of the curve.

Table F.1: Comparison of threshold schedules. The cosine schedule with $0.55 \rightarrow 0.40$ achieves the best result.

| Range | Method | mAP |
|---|---|---|
| $0.55 \rightarrow 0.40$ | Cosine | **53.2** |
| $0.80 \rightarrow 0.40$ | Cosine | 45.7 |
| $0.55 \rightarrow 0.20$ | Cosine | 51.2 |
| $0.80 \rightarrow 0.20$ | Cosine | 47.0 |
| 0.50 | Fixed | 50.9 |
| 0.40 | Fixed | 48.9 |
| 0.55 | Fixed | 50.8 |
| $0.55 \rightarrow 0.40$ | Exponential | 50.5 |
| $0.55 \rightarrow 0.40$ | Sigmoid | 48.8 |

**Analysis.** Table F.1 summarizes the results (mAP@50) under different scheduling strategies and thresholds. The comparison shows that cosine scheduling consistently outperforms fixed thresholds. In particular, the best cosine setting ($0.55 \rightarrow 0.40$) achieves **53.2** mAP, which surpasses the best fixed threshold (0.50, 50.9) by a margin of **+2.3**. This demonstrates the advantage of progressively relaxing the confidence threshold rather than keeping it static throughout training.

The choice of threshold bounds also has a clear impact. When $\tau_{\max}$ is set too high (e.g., 0.8), pseudo-label coverage becomes overly restrictive in the early stages, leading to poor adaptation results (45.7/47.0). Conversely, when $\tau_{\min}$ is set too low (e.g., 0.2), the model admits too many noisy pseudo labels in later stages, which also degrades performance (51.2/47.0). Moderate ranges, such as $0.55 \rightarrow 0.40$, provide the most balanced trade-off between early precision and late recall.

When comparing different dynamic strategies under the same threshold range ($0.55 \rightarrow 0.40$), the cosine schedule again proves most effective (53.2), outperforming exponential (50.5) and sigmoid (48.8). The key reason is that cosine decay (Eq. F.2) introduces a smooth and symmetric relaxation, which avoids the overly aggressive early drop of exponential decay (Eq. F.3) and the saturation effects that slow down sigmoid adjustment (Eq. F.4).

From a practical perspective, in noisier domains, slightly higher $\tau_{\min}$ may be beneficial to suppress noise, whereas in cleaner domains, reducing $\tau_{\min}$ can further expand recall without sacrificing stability.

## G  ABLATION: SLOT NUMBER AND ARCHITECTURE

To further justify our choice of **HSA**, we conduct both qualitative and quantitative ablations against vanilla Slot Attention (SA) under different slot numbers. Figure G.1 visualizes the per-slot masks; Table G.1 reports the detection performance (mAP@50). Unless otherwise stated, the training schedule, backbone, and detector remain fixed; only the slot mechanism differs.

**SA (num_slots = 5).** With only 5 slots, SA roughly separates foreground from background but remains too coarse: fine-grained structures (e.g., cars, pedestrians) are merged into large regions, limiting the discriminability of object-level priors. This indicates that a small slot budget is insufficient for complex scenes.

**SA (num_slots = 25).** Simply increasing the slot count to 25 does not yield finer decomposition. Instead, we observe severe *striping artifacts*, where multiple slots **collapse** into redundant, nearly vertical partitions, reflecting training instability under large slot numbers on real-world data. Notably, although we set 25 slots, the masks are produced via competition across slots. Collapsed or low-confidence slots receive near-zero weights and thus become effectively inactive or overlap with dominant slots. The final argmax visualization therefore does not exhibit 25 distinct segments.

**HSA (num_slots = $5 \times 5$).** **HSA** performs coarse-to-fine decomposition: five stable coarse slots are first established and then refined into 25 fine slots, avoiding the collapse of vanilla SA. As shown

in Figure G.1, **HSA** generates spatially coherent, object-aligned partitions that achieve both stability and granularity.

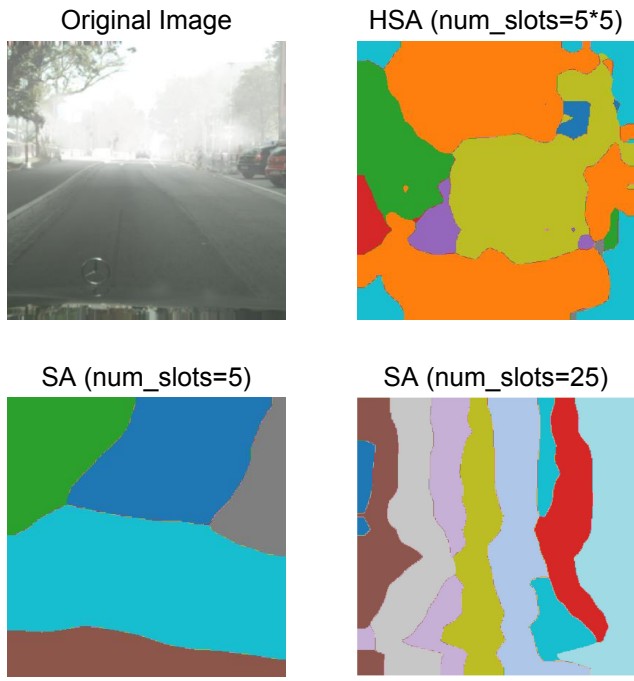

Figure G.1: Visualization of mask decomposition under different configurations. Top-left: Original image. Top-right: HSA ($5 \times 5$ slots). Bottom-left: SA (5 slots). Bottom-right: SA (25 slots). HSA achieves both stability and granularity, while vanilla SA either under-segments or collapses.

Table G.1: Quantitative ablation on slot configurations (mAP@50). HSA consistently outperforms vanilla SA under both small and large slot settings.

| Method | mAP |
|---|---|
| SA (num_slots = 5) | 50.6 |
| SA (num_slots = 25) | 50.2 |
| HSA ($5 \times 5$ slots) | **53.2** |

**Analysis.** From Table G.1, we observe the following: (i) **SA-5** attains 50.6 mAP, matching the qualitative observation that coarse segmentation captures only parts of the foreground. (ii) **SA-25** fails to improve due to slot collapse and redundancy—some slots become inactive and provide little additional capacity. (iii) **HSA** ($5 \times 5$) achieves 53.2 mAP, validating that hierarchical decomposition delivers stable training and effective fine-grained priors.

Therefore, **HSA** is superior to vanilla SA: (i) it alleviates under-segmentation with few slots; (ii) it avoids collapse when the slot count is large; and (iii) it provides stable, fine-grained object-level priors that benefit source-free adaptation.

## H  VISUALIZATION OF THE SLOT-BASED DECOMPOSITION

**Visualizations of the slots.** As shown in Figure H.1, across diverse driving scenes in the BDD100K, the slots consistently partition the image into spatially coherent regions that roughly align with objects or large structures (e.g., cars, road surfaces, buildings, sky, vegetation). Importantly, this behavior emerges without any pixel-level supervision: **HSA** is trained purely with a reconstruction objective and competitive assignment among slots. As a result, the decomposition is

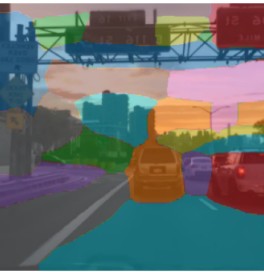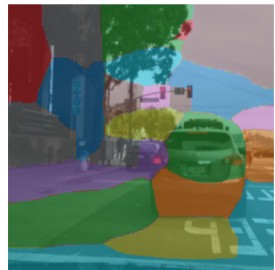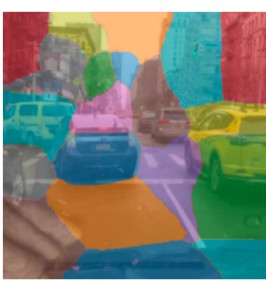

Figure H.1: Additional visualizations of the slots.

not intended to match semantic segmentation perfectly. This is main reason why some regions can span multiple semantic categories, and some small objects may be merged with nearby background. This phenomenon is expected in unsupervised slot-based models and has also been observed in prior object-centric learning works (Locatello et al., 2020; Seitzer et al., 2022).

What HSA provides, therefore, is not a semantic segmentation, but a rich structural prior: the model is encouraged to represent the scene using a small number of object-like regions, whose boundaries tend to follow strong edges and instance contours. This prior is exactly what our theoretical analysis exploits to reduce background variance and extract domain-invariant features. To further enhance the semantic consistency of these structural slots, we introduce the **CGSC** module, which aligns slot prototypes with class prototypes during adaptation. **HSA** focuses on how the scene is decomposed, while **CGSC** focuses on which class each region should correspond to.

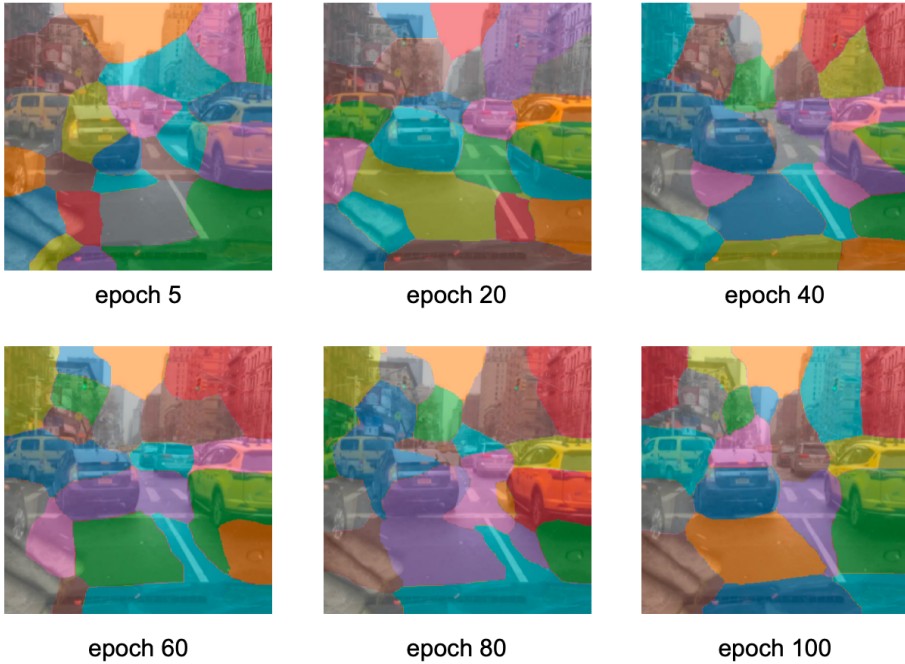

Figure H.2: Visualization of the decomposition process

**Visualization of the decomposition process.** To make the slot decomposition process more intuitive, we visualize the masks produced by **HSA** at different training epochs on the same target image from the BDD100K, as shown in Figure H.2. At the beginning of training (epoch 5), the slots appear as irregular, fragmented patches that mix cars, road, and buildings, indicating that the model

has not yet learned meaningful groups. As training proceeds (epochs 20–80), the decomposition becomes progressively more structured: slot boundaries increasingly follow strong image edges and large regions such as road, sky, and building facades are covered by a few coherent slots, while other slots start to focus on car-shaped regions. By epoch 100, the slots stabilize into a consistent partition of the scene, where a few slots cover the road, several specialize on individual cars or rows of cars, and others capture background structures like buildings and sky.

This evolution from noisy, mixed patches to clean, object-like regions illustrates how **HSA** progressively learns an object-centric decomposition without any pixel-level supervision. The visualization supports our claim that **HSA** provides meaningful structural priors as the slots specialize to coherent regions that roughly correspond to objects or large parts over the course of training.

## I    EMPIRICAL VALIDATION OF THEORETICAL ANALYSIS

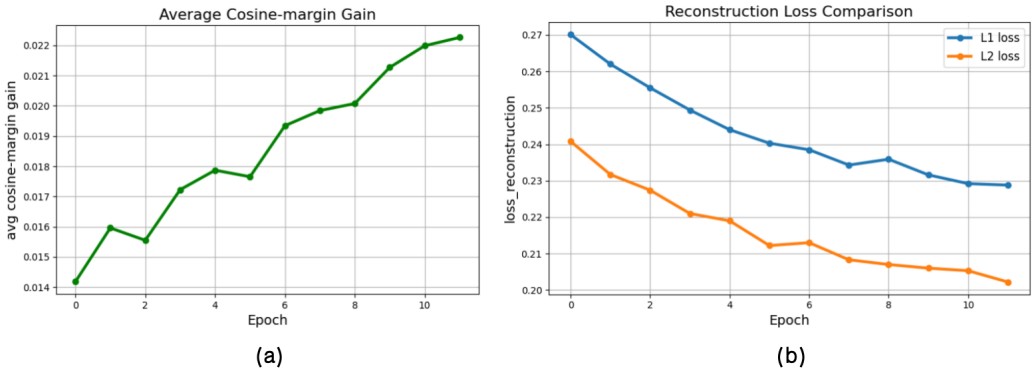

Figure I.1: Empirical validation of cosine-margin gain and reconstruction consistency

To further substantiate our theoretical analysis, we now provide training dynamics on the Cityscapes→Foggy Cityscapes. The reported cosine-margin gain is computed exactly according to the definition in Eq.(A.4) of the appendix, and for the reconstruction consistency of **HSA** we additionally test two reconstruction objectives (L1 and L2) to verify that the observed convergence behavior is not tied to a specific metric.

As shown in Fig.I.1(a), the average cosine-margin gain increases steadily during training, rising from about 0.014 at the start to above 0.022 at the end. This monotonic upward trend empirically supports our theoretical claim that **CGSC** enlarges the cosine margin between same-class and cross-class prototype–slot similarities, which is a key condition in our risk-descent analysis.

Meanwhile, Fig.I.1(b) shows that the reconstruction loss decreases consistently over epochs under both L1 and L2 formulations, indicating progressively better and more stable slot-based decomposition, which we refer to as reconstruction consistency. Importantly, both reconstruction objectives lead to the same qualitative trend of steady convergence, suggesting that our approach is robust to the specific choice of reconstruction metric. Quantitatively, L2 reconstruction yields a slightly higher final detection performance (53.2 vs. 52.5 mAP), so we adopt L2 as the default in our framework.

## J    IMPACT OF COCO PRETRAINING FOR HSA

To quantify the impact of COCO self-supervised pretraining, we have conducted experiments on the Cityscapes→Foggy where **HSA** is initialized randomly (no COCO pretraining), with all other experimental settings identical to our approach. The results are shown in Tab.J.1.

From Tab. J.1 we observe that, even without COCO pretraining, adding **HSA** and **CGSC** already brings large gains over the baseline: **HSA** alone improves mAP from 46.2 to 51.2 (+5.0), and the full **CGSA** further raises it to 52.7 (+6.5). This indicates that **HSA** can be learned purely from the in-domain data (source and target) and still provides strong object-centric priors for adaptation. COCO

Table J.1: Ablation Study of COCO pretraining for HSA.

| Setting | COCO pretraining for HSA | mAP |
|---------|--------------------------|-----|
| Source only | – | 46.2 |
| w/ HSA | × | 51.2 |
| w/ HSA & CGSC (CGSA) | × | 52.7 |
| w/ HSA | ✓ | 51.8 |
| w/ HSA & CGSC (CGSA) | ✓ | 53.2 |

self-supervised pretraining gives an additional but moderate boost: the mAP of **HSA** increases from 51.2 to 51.8 (+0.6), and that of **CGSA** from 52.7 to 53.2 (+0.5).

In other words, only a small fraction of the total improvement over the source-only baseline can be attributed to COCO pretraining, whereas the majority of the improvement stems from the proposed mechanism itself. Therefore, COCO pretraining is helpful but not critical.

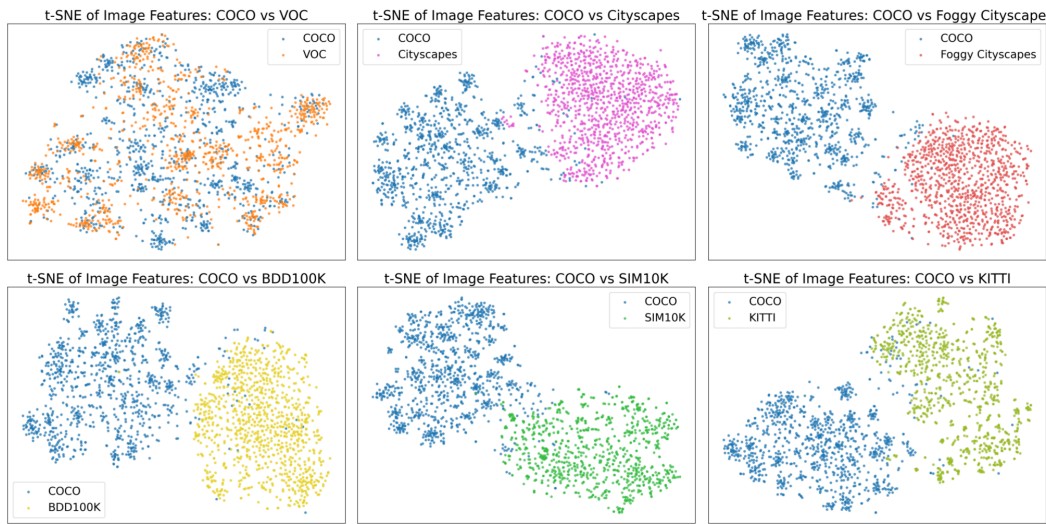

Figure J.1: Visualization of distribution differences between COCO and all datasets used in our experiments.

Besides, to further explore how COCO pretraining affects the generalization capability of **HSA**, we visualize the feature distributions of COCO and all datasets used in our experiments, as shown in Fig. J.1. To ensure a fair comparison, we use the same frozen DINO backbone as in **HSA** pretraining, we extract global image features and apply t-SNE for dimensionality reduction. As a reference case, we also plot COCO vs Pascal VOC (first row, first column). These two classic datasets are both generic web images with centrally framed, prominent objects, and their t-SNE embeddings significantly overlap, confirming that their distributions are relatively similar.

In contrast, when we compare COCO with Cityscapes, Foggy-Cityscapes, BDD100K, Sim10K, and KITTI, the two feature distributions are clearly separated in every plot. These datasets are all related to driving scenarios but differ substantially from COCO's general, static object photographs. Nevertheless, our COCO-pretrained **HSA** consistently improves performance across all these benchmarks in the main experiments. This strongly suggests that the COCO-pretrained **HSA** generalizes well, as evidenced by its ability to transfer the object-centric structural prior learned from generic web images to very different driving domains, rather than overfitting to COCO-specific statistics.

## K  COMPUTATIONAL OVERHEAD

To analyze the computational overhead of **HSA**, we report the parameters, per-iteration processing time, and mAP on two different base detectors: a very lightweight RT-DETR (Zhao et al., 2024)

Table K.1: Analysis of computational overhead. Results are measured on the Cityscapes → Foggy-Cityscapes setting.

| Base Detector | Component | Parameters | Time | mAP |
|---|---|---|---|---|
| RT-DETR (Zhao et al., 2024) | – | 43M | 269ms | 46.2 |
| | + HSA | 63M | 343ms | 51.8 |
| | Δ% | 46.5% | 27.5% | 12.1% |
| FocalNet-DINO-DETR (Yang et al., 2022) | – | 228M | 780ms | 43.8 |
| | + HSA | 251M | 809ms | 55.0 |
| | Δ% | 10.1% | 3.7% | 25.6% |

and a large FocalNet-DINO detector (Yang et al., 2022). As shown in Table K.1, we can observe that our method achieves significant performance gains in both detection frameworks. Although the introduction of **HSA** increases parameters by 46.5% and training time by 27.5% on the lightweight RT-DETR, it is essential to note that RT-DETR is designed as an extremely lightweight DETR variant for real-time detection, comparable to YOLO-style models. Consequently, any non-trivial module inevitably results in a relatively large percentage increase in parameters. In contrast, on the much heavier FocalNet-DINO, the relative overhead is much smaller (10.1% parameters, 3.72% training time), yielding a remarkable +25.6% mAP improvement. This makes the computational cost of HSA negligible. These results demonstrate that HSA introduces a controllable and reasonable computational overhead.

## L    ETHICS STATEMENT

This work focuses on proposing and analyzing methodological advances, with all experiments conducted on publicly available datasets and open-source models. No private or sensitive data are involved in this study. Our contribution lies in understanding the underlying mechanisms and improving model performance, rather than deploying downstream applications that may raise ethical or societal concerns. Therefore, this work does not pose direct risks of misuse.

## M    REPRODUCIBILITY STATEMENT

We have made significant efforts to ensure the reproducibility of our work. The main paper describes the model architecture and training procedure in detail (Section 3). Additional implementation details and hyperparameters are provided in the Appendix B. The pseudocode of the main algorithm is presented in the Appendix E. The code is released at https://github.com/Michael-McQueen/CGSA.

## N    THE USE OF LARGE LANGUAGE MODELS (LLMS)

The authors used ChatGPT-5 to solely polish the readability and grammatical accuracy of selected passages after the initial draft was completed.

