# OpenReview forum: "CGSA: Class-Guided Slot-Aware Adaptation for Source-Free Object Detection"
_ICLR.cc/2026/Conference — ICLR 2026 Poster_

### Official Review · Reviewer_GFAW · 2025-10-28

**Soundness:** 3
**Presentation:** 3
**Contribution:** 3
**Rating:** 6
**Confidence:** 4

**Summary:**

The paper proposes CGSA, a novel source-free domain adaptive object detection framework that integrates object-centric learning into DETR-based models. It introduces two modules: Hierarchical Slot Awareness, which decomposes images into slot-based structural priors, and Class-Guided Slot Contrast, which aligns slots with class semantics through contrastive learning. The approach enables domain-invariant adaptation without source data. Experiments on multiple benchmarks show consistent and significant improvements over existing methods, supported by theoretical risk descent analysis.

**Strengths:**

1. The paper introduces CGSA, the first framework to combine Object-Centric Learning with Source-Free Domain Adaptive Object Detection, and integrates slot-based structural priors and class-guided contrastive alignment reasonably to achieve domain-invariant adaptation.
2. Extensive experiments show that CGSA significantly outperforms prior methods, and the authors also provide theoretical analysis demonstrating risk reduction and stable convergence during adaptation.

**Weaknesses:**

1. Nevertheless, the proposed method remains highly dependent on the DETR framework, showing limited generalizability. Evaluating its effectiveness on other architectures, such as Faster R-CNN, is still desireable.
2. There is no intuitive visualization or analysis of slot decomposition process.
3. The risk analysis is mathematically sound but lacks empirical validation of key variables in authors' description, e.g., cosine margin gain or reconstruction consistency.

**Questions:**

1. How sensitive is CGSA to the number of slots (n=5) or slot hierarchy depth?
2. What is the computational overhead compared to standard DETR?

---

> ### Author Response · Authors · 2025-11-22
> **Reply from the Authors (Part 1)**
>
> ## 1. Regarding the generalizability of the basic detector, such as Faster R-CNN
>
> Thank you for this insightful comment. However, we would like to highlight that our method is deliberately designed around the DETR architecture and its query tokens. The motivation behind this choice is two-fold:
>
> * ​**Architectural perspective**​: DETR represents a paradigm shift in object detection by eliminating hand-crafted components such as anchors and non-maximum suppression, providing a more unified and end-to-end learning pipeline. **This design naturally complements our approach, which builds upon the object query mechanism to enable more efficient representation learning**.
> * ​**Alignment with recent advances**​: Recent advances in domain adaptive object detection, including FRANCK [1] and TITAN [2], have increasingly adopted DETR-based detectors, underscoring a clear and growing trend within the research community. By aligning with this modern trajectory, our work naturally integrates with contemporary SFOD advancements and facilitates more meaningful comparisons and extensions in future research [1][2].
>
> Here we provide more details:
>
> In DETR, a fixed number of object queries form a permutation-invariant set. Each query is intended to represent one potential object, and the final predictions are obtained via bipartite matching between queries and ground-truth boxes. This set-prediction view naturally matches our object-centric design:
>
> - HSA first decomposes the global encoder feature map into a small set of slots, each summarizing an object-like region.
> - These slots are then injected into the object queries, providing them with structured, object-level visual priors before decoding.
> - Our theoretical analysis is built on this slot–query coupling, where each query interacts with a small number of slots that encode object-centric structural priors.
>
> In contrast, Faster R-CNN represents objects via local RoI features cropped from thousands of region proposals. There is no fixed set of global queries, no one-to-one set prediction, and proposals are heavily tied to anchor design and NMS. Mapping our slots onto this pipeline is inherently awkward:
>
> - Slots represent globally assigned object-level regions, whereas RoIs are local crops determined by RPN. The only feasible option is to fuse the slots with all RoIs, which largely defeats the purpose of competitive global decomposition.
> - The proposal-based pipeline already provides its own object hypotheses. Injecting another global decomposition on top tends to be redundant and has much weaker correspondence with the final classification/regression heads than in DETR, where every prediction must go through query–slot interaction.
>
> For these reasons, we view CGSA as conceptually DETR-specific: it exploits the global set of object queries and the Hungarian matching structure, which are absent in Faster R-CNN.
>
> While Faster R-CNN is a well-established detector, recent work in domain adaptive object detection has shown that DETR-based detectors are highly competitive and worth studying in their own right. Several UDA methods are built specifically for detection transformers. SFA[3] firstly aligns encoder/decoder token sequences in Deformable DETR for cross-domain detection. AQT[4] introduces adversarial query transformers that perform adversarial alignment directly on DETR feature tokens. MRT[5] and DATR[6] further design teacher–student retraining and class-wise prototype alignment tailored for DETR-style detectors, achieving strong DAOD performance.
>
> In the SF-DAOD setting, TITAN[2] and FRANCK[1] both explicitly choose DETR as the base detector and design query-centric mechanisms (query-token adversarial learning, query-fused feature reweighting and distillation) to adapt DETR to new domains. These methods already report state-of-the-art SF-DAOD results on DETR, demonstrating that exploring DETR architectures in this setting is not only fair but also desirable.
>
> Our work follows this line of research: we adopt DETR as the base detector but introduce a new object-centric perspective via HSA and CGSC. On top of a strong DETR baseline, our method achieves new SOTA performance among SF-DAOD methods, suggesting that object-centric priors are a promising direction for domain adaptation.

---

> > ### Author Response · Authors · 2025-11-22
> > **Reply from the Authors (Part 2)**
> >
> > ## 2. Intuitive visualization of the slot decomposition process
> >
> > Thank you for this useful suggestion. To make the slot decomposition process more intuitive, we visualize the masks produced by HSA at different training epochs on the same target image from the BDD100k, as shown in Figure I.1 in Appendix I.
> >
> > At the beginning of training (epoch 5), the slots appear as irregular, fragmented patches that mix cars, road, and buildings, indicating that the model has not yet learned meaningful groups. As training proceeds (epochs 20–80), the decomposition becomes progressively more structured: slot boundaries increasingly follow strong image edges and large regions such as road, sky, and building facades are covered by a few coherent slots, while other slots start to focus on car-shaped regions. By epoch 100, the slots stabilize into a consistent partition of the scene, where a few slots cover the road, several specialize on individual cars or rows of cars, and others capture background structures like buildings and sky.
> >
> > This evolution from noisy, mixed patches to clean, object-like regions illustrates how HSA progressively learns an object-centric decomposition without any pixel-level supervision. The visualization supports our claim that HSA provides meaningful structural priors as the slots specialize to coherent regions that roughly correspond to objects or large parts over the course of training.
> >
> >
> > ## 3. Empirical validation of cosine-margin gain and reconstruction consistency
> >
> > We thank the reviewer for this insightful comment. Following the reviewer’s suggestion, we now provide training dynamics on the Cityscapes→Foggy Cityscapes. The reported cosine-margin gain is computed exactly according to the definition in Eq.(A.4) of the appendix, and for the reconstruction consistency of HSA, we additionally test two reconstruction objectives (L1 and L2) to verify that the observed convergence behavior is not tied to a specific metric.
> >
> > As shown in Figure J.1(a) in Appendix J, the average cosine-margin gain increases steadily during training, rising from about 0.014 at the start to above 0.022 at the end. This monotonic upward trend empirically supports our theoretical claim that CGSC enlarges the cosine margin between same-class and cross-class prototype–slot similarities, which is a key condition in our risk-descent analysis. This conclusion is further supported qualitatively by Figure 5 in the main paper.
> >
> > Meanwhile, Figure J.1(b) in Appendix J shows that the reconstruction loss of HSA decreases consistently over epochs under both L1 and L2 formulations, indicating progressively better and more stable slot-based decomposition, which we refer to as reconstruction consistency. Importantly, both reconstruction objectives lead to the same qualitative trend of steady convergence, suggesting that our method is robust to the specific choice of reconstruction metric. Quantitatively, L2 reconstruction yields a slightly higher final detection performance (53.2 vs. 52.5 mAP), so we adopt L2 as the default in CGSA.
> >
> > Overall, the simultaneous increase of cosine margin and decrease of reconstruction loss provides direct empirical validation of the key variables used in our theoretical analysis, strengthening the connection between our risk-descent theory and practical training dynamics.

---

> > > ### Author Response · Authors · 2025-11-22
> > > **Reply from the Authors (Part 3)**
> > >
> > > ## 4. Sensitivity to slot number and hierarchy depth
> > >
> > > We thank the reviewer for this question and provide a parameter sensitivity study in Table R4.1. Here “Depth” denotes the hierarchy depth of HSA, and “Number” is the slot count per level. So the total number of fine slots is $ Number^{Depth} $.
> > >
> > > **Table R4.1** Parameter Sensitivity Analysis of HSA
> > >
> > > |Depth   |  Number   | Total Slot Numbers  | mAP
> > > |:------:|:----:     |:----:|:-----:|
> > > | 2      |2    | 4 | 50.9  |
> > > | 2      |4    | 16 | 52.9  |
> > > | 2      |5    | 25 | **53.2**  |
> > > | 2      |6    | 36 | 52.6  |
> > > | 2      |8    | 64 | 52.5  |
> > > | 2      |10   | 100 | 51.2  |
> > > | 3      |2    | 8 | 50.8  |
> > > | 3      |3    | 27 | 50.3  |
> > > | 3      |4    | 64 | 48.7  |
> > >
> > > When we use two-level HSA (depth=2), the performance is not highly sensitive to the slot number. Across a wide range of $number \in \{2,4,5,6,8,10\}$, mAP stays within a relatively small and acceptable band , showing robustness to this hyperparameter. The observed trend is both theoretically and intuitively justified, exhibiting a pattern that is "low at both ends, high in the middle", where very small number (too few slots, e.g., 4 total slots) cannot capture sufficient object structure, while very large numbers(too many slots, e.g., 64 and 100 total slots) dilutes object-centric grouping and tends to fragment objects, thus reducing the usefulness of the structural prior. The best trade-off is achieved around number=5 (25 fine slots), which gives the highest mAP of 53.2.
> > >
> > > Increasing the hierarchy depth to 3 leads to a clear performance drop. We attribute this to the unsupervised nature of slot attention, since HSA is trained only with self-supervision, and deeper hierarchies weaken the reconstruction constraint at each level and accumulate decomposition errors, resulting in lower-quality slots for adaptation.
> > >
> > > Finally, when Depth = 1, HSA degenerates to standard Slot Attention. As discussed in Appendix G, training becomes unstable and often collapses when number is large, which motivates our hierarchical design. Therefore, considering both robustness and performance, we choose depth = 2 and number=5 as our default setting.
> > >
> > > ## 5. Computational overhead of HSA
> > >
> > > Thank you for this insightful comment. To address the reviewer’s concern, we report the parameters, per-iteration processing time[1], and mAP with and without HSA on two different base detectors: a very lightweight RT-DETR[7] and a large FocalNet-DINO detector[8]. Results are measured on the Cityscapes → Foggy-Cityscapes setting.
> > >
> > > **Table R4.2** Ablation on computing analysis
> > >
> > > | Base Detector        | Component     | Parameters | Time     | mAP |
> > > |:-------:             |:-------------:|:----------:|:------:    |:---:|
> > > | RT-DETR[7]           | -             | 43M        | 269ms      | 46.2    |
> > > |                      | + HSA         | 63M        | 343ms      | 51.8    |
> > > |                      | $\Delta$%     | 46.5%      | 27.5%      | 12.1%   |
> > > | FocalNet-DINO-DETR[8]| -             | 228M       | 780ms      | 43.8    |
> > > |                      | + HSA         | 251M       | 809ms      | 55.0    |
> > > |                      | $\Delta$%     | 10.1%      | 3.7%      | 25.6%    |
> > >
> > > As shown in Table R4.2, we can observe that our method achieves significant performance gains in both detection frameworks. Although the introduction of HSA increases parameters by 46.5% and training time by 27.5% on the lightweight RT-DETR, it is essential to note that RT-DETR is designed as an extremely lightweight DETR variant for real-time detection, comparable to YOLO-style models. Consequently, any non-trivial module inevitably results in a relatively large percentage increase in parameters. In contrast, on the much heavier FocalNet-DINO, the relative overhead is much smaller (10.1% parameters, 3.72% training time), yielding a remarkable +25.6% mAP improvement. This makes the computational cost of HSA negligible. These results demonstrate that HSA introduces a controllable and reasonable computational overhead.
> > >
> > > We sincerely thank the reviewer for the time and effort spent on our paper. We believe our responses have thoroughly addressed the concerns raised, and we are happy to engage in any further discussion that may be needed.

---

> > > > ### Author Response · Authors · 2025-11-22
> > > > **Reply from the Authors (Part 4)**
> > > >
> > > > **Reference**
> > > >
> > > > 1. Yao H, Zhao S, Lu S, et al. Source-Free Object Detection with Detection Transformer[J]. IEEE Transactions on Image Processing, 2025.
> > > > 2. Ashraf T, Bashir J. TITAN: Query-Token based Domain Adaptive Adversarial Learning[J]. arXiv preprint arXiv:2506.21484, 2025.
> > > > 3. Li X, Chen W, Xie D, et al. A free lunch for unsupervised domain adaptive object detection without source data[C]//Proceedings of the AAAI Conference on Artificial Intelligence. 2021, 35(10): 8474-8481.
> > > > 4. Huang W J, Lu Y L, Lin S Y, et al. AQT: Adversarial Query Transformers for Domain Adaptive Object Detection[C]//IJCAI. 2022: 972-979.
> > > > 5. Zhao Z, Wei S, Chen Q, et al. Masked retraining teacher-student framework for domain adaptive object detection[C]//Proceedings of the IEEE/CVF International Conference on Computer Vision. 2023: 19039-19049.
> > > > 6. Chen L, Han J, Wang Y. Datr: Unsupervised domain adaptive detection transformer with dataset-level adaptation and prototypical alignment[J]. IEEE Transactions on Image Processing, 2025.
> > > > 7. Zhao Y, Lv W, Xu S, et al. Detrs beat yolos on real-time object detection[C]//Proceedings of the IEEE/CVF conference on computer vision and pattern recognition. 2024: 16965-16974.
> > > > 8. Yang J, Li C, Dai X, et al. Focal modulation networks[J]. Advances in Neural Information Processing Systems, 2022, 35: 4203-4217.

---

> > > > > ### Author Response · Authors · 2025-11-25
> > > > > **A kind reminder from the authors**
> > > > >
> > > > > Dear Reviewer GFAW,
> > > > >
> > > > > As the discussion window is coming to a close in just over a week, we would like to gently inquire whether our replies have fully resolved your concerns. If not, we would greatly appreciate the opportunity to continue the dialogue and will do our best to respond to any additional questions. If you believe the concerns have been addressed, we would be grateful if you could consider increasing your score accordingly.
> > > > >
> > > > > Thank you very much!
> > > > >
> > > > > The Authors of submission 17563

---

### Official Review · Reviewer_5pPM · 2025-10-30

**Soundness:** 3
**Presentation:** 3
**Contribution:** 3
**Rating:** 6
**Confidence:** 4

**Summary:**

This paper proposes integrating object-centric learning into DETR-based detectors for source-free domain adaptive object detection. Specifically, the Hierarchical Slot Awareness (HAS) module decomposes an image into a set of slots, which are then fused with queries to inject object-level visual priors into the object decoder. Additionally, a Class-Guided Slot Contrast (CGSC) mechanism is introduced to guide the slots toward domain-invariant yet class-relevant object features by leveraging contrastive learning with class prototypes. Extensive experiments on multiple datasets demonstrate the effectiveness of the proposed method.

**Strengths:**

1. The method achieves state-of-the-art performance across multiple datasets.

2. The authors provide a solid theoretical analysis explaining why slot-based features can offer domain-invariant priors.

**Weaknesses:**

1. The HAS module introduces additional parameters and computation overhead. It would be beneficial to provide a comparison of speed and parameter size before and after adding HAS.

2. The authors did not provide the performance of the source-only model after adding HAS. This omission makes it unclear whether the performance improvement stems from a stronger source-only model or from HAS enhancing the adaptation capability.

3. HAS requires self-supervised pretraining on the COCO dataset. It would be useful to discuss the importance of this pretraining step.

**Questions:**

1. Why is CGSC not used during training on the source data?
2. Can this method be applied to source-available domain adaptive object detection (DAOD)?

---

> ### Author Response · Authors · 2025-11-22
> **Reply from the Authors (Part 1)**
>
> ## 1. Computational overhead of HSA
>
> Thank you for this insightful comment. To address the reviewer’s concern, we report the parameters, per-iteration processing time[1], and mAP with and without HSA on two different base detectors: a very lightweight RT-DETR[2] and a large FocalNet-DINO detector[3]. Results are measured on the Cityscapes → Foggy-Cityscapes setting.
>
> **Table R3.1** Ablation on computing analysis
>
> | Base Detector        | Component     | Parameters | Time     | mAP |
> |:-------:             |:-------------:|:----------:|:------:    |:---:|
> | RT-DETR[2]           | -             | 43M        | 269ms      | 46.2    |
> |                      | + HSA         | 63M        | 343ms      | 51.8    |
> |                      | $\Delta$%     | 46.5%      | 27.5%      | 12.1%   |
> | FocalNet-DINO-DETR[3]| -             | 228M       | 780ms      | 43.8    |
> |                      | + HSA         | 251M       | 809ms      | 55.0    |
> |                      | $\Delta$%     | 10.1%      | 3.7%      | 25.6%    |
>
> As shown in Table R3.1, we can observe that our method achieves significant performance gains in both detection frameworks. Although the introduction of HSA increases parameters by 46.5% and training time by 27.5% on the lightweight RT-DETR, it is essential to note that RT-DETR is designed as an extremely lightweight DETR variant for real-time detection, comparable to YOLO-style models. Consequently, any non-trivial module inevitably results in a relatively large percentage increase in parameters. In contrast, on the much heavier FocalNet-DINO, the relative overhead is much smaller (10.1% parameters, 3.72% training time), yielding a remarkable +25.6% mAP improvement. This makes the computational cost of HSA negligible. These results demonstrate that HSA introduces a controllable and reasonable computational overhead.
>
> ## 2. Does HSA only strengthen the source-only model?
>
> We deeply appreciate the reviewer’s comment about whether the gains come merely from a stronger source-only detector. To clarify this, we additionally trained a source-only model with HSA (directly adaptation) on the Cityscapes→Foggy Cityscapes. The results are shown in Table R3.2. "Source only" indicates that "Target testing without target data training". "Adaptation" indicates that "Target testing with target data training".
>
> **Table R3.2** Ablation Study on Source-Free setting.
>
> | Setting            | HSA | CGSC | mAP |
> | :---------------:  | :-: | :--: | :----: |
> | Source only        |  ✗  |   ✗  |  46.2 |
> | Source only        |  ✓  |   ✗  |  47.3 |
> | Adaptation         |  ✓  |   ✗  |  51.8 |
> | Adaptation(ours)   |  ✓  |   ✓  |  53.2 |
>
> When HSA is used only in source-only training, the performance improves **from 46.2 to 47.3 mAP (+1.1 mAP)**. This shows that HSA does slightly strengthen the base detector, but the effect on its own is modest. Once we enable source-free adaptation on the target domain with HSA, the performance jumps further from 46.2 to 51.8 mAP (+5.6). This ablation supports our claim that HSA is not just making the baseline marginally stronger, but is primarily useful as an object-centric structural prior for domain adaptation, especially when combined with our class-guided contrastive alignment.

---

> > ### Author Response · Authors · 2025-11-22
> > **Reply from the Authors (Part 2)**
> >
> > ## 3. Importance of COCO self-supervised pretraining for HSA
> >
> > We thank the reviewer for raising this point. To quantify the impact of COCO self-supervised pretraining, we have conducted experiments on the Cityscapes→Foggy Cityscapes, where HSA is initialized randomly (no COCO pretraining), with all other experimental settings identical to our approach. The results are shown in Table R3.3.
> >
> > **Table R3.3** Ablation Study of COCO pretraining for HSA. "Source Only" indicates that "Target testing without target data training and using only the base detector".
> >
> > | Setting                   | COCO pretraining for HSA | mAP    |
> > | :-----------------------: | :----------------------: | :--------: |
> > | Source only               | –                        |  46.2 |
> > | w/ HSA                    | ✗                       |  51.2 |
> > | w/ HSA & CGSC (CGSA)            | ✗                        |  52.7 |
> > | w/ HSA                    | ✓                        |  51.8 |
> > | w/ HSA & CGSC (CGSA)            | ✓                        |  53.2 |
> >
> > From Table R3.3, we observe that, even without COCO pretraining, adding HSA and CGSC already brings large gains over the baseline: HSA alone improves mAP from 46.2 to 51.2 (+5.0), and the full CGSA further raises it to 52.7 (+6.5). This indicates that HSA can be learned purely from the in-domain data (source and target) and still provides strong object-centric priors for adaptation. **COCO self-supervised pretraining gives an additional but moderate boost: the mAP of HSA increases from 51.2 to 51.8 (+0.6), and that of CGSA from 52.7 to 53.2 (+0.5).**
> >
> > In other words, only a small fraction of the total improvement over the source-only baseline can be attributed to COCO pretraining, whereas the majority of the improvement stems from the HSA/CGSC mechanism itself. Therefore, COCO pretraining is helpful but not critical. It offers a slightly better initialization, leading to smoother optimization and a small performance gain, but the core benefits of our approach do not rely on COCO datasets, and HSA remains effective when trained from scratch within our source-free adaptation pipeline.
> >
> > ## 4. Why is CGSC not applied on the source data
> >
> > Thank you for this thoughtful comment. To investigate whether CGSC should also be used during supervised training on the source domain, we conducted additional experiments on Cityscapes→Foggy Cityscapes, as shown in Table R3.4.
> >
> > **Table R3.4** Effect of applying CGSC on source-only setting. "Source only" indicates that "Target testing without target data training". "Adaptation" indicates that "Target testing with target data training".
> >
> > | Setting            | HSA | CGSC | mAP |
> > | :---------------:  | :-: | :--: | :----: |
> > | Source only        |  ✗  |   ✗  |  46.2 |
> > | Source only        |  ✓  |   ✓  |  46.6 |
> > | Adaptation         |  ✓  |   ✓  |  52.6 |
> > | Source only        |  ✓  |   ✗  |  47.3 |
> > | Adaptation(ours)   |  ✓  |   ✓  |  53.2 |
> >
> > When the model uses only HSA on the source domain, performance improves to 47.3 mAP (+1.1), indicating that the structural prior itself contributes more significantly to strengthening the detector. Building upon this, adding CGSC on the source domain results in a slight performance drop (-0.7), suggesting that excessively emphasizing contrastive constraints on the already well-supervised source domain may lead to overfitting or undermine the benefits of HSA. Besides, using CGSC on both source and target domains yields an mAP of 52.6, while applying it solely on the target domain (our setting) further boosts the mAP to 53.2. This demonstrates that the primary value of CGSC lies in enforcing class consistency and domain invariance for the target slots, rather than redundantly providing similar information on the source domain. Simultaneously, its exclusive use on the target domain avoids the minor performance degradation associated with source-domain overfitting.

---

> > > ### Author Response · Authors · 2025-11-22
> > > **Reply from the Authors (Part 3)**
> > >
> > > ## 5. Applicability of CGSA to source-available DAOD
> > >
> > > We thank the reviewer for raising this question about the generality of our framework. To investigate this, we directly apply CGSA to the source-available DAOD setting, without any modification.
> > >
> > > **Table R3.5** Results on source available setting.
> > >
> > > |SF           | C→F  | C→BDD | S→C | K→C |
> > > |:-----------:|:----:|:-----:|:---:| :---:|
> > > | ✗    | 53.9 | 53.1  | 69.8| 61.2 |
> > > | ✓    | 53.2 | 53.0  | 67.7| 60.8 |
> > >
> > > As shown in Table R3.5, when both source and target data are accessible (SF ✗), CGSA achieves slightly higher mAP than in the source-free setting on all four benchmarks. The gains are relatively modest (e.g., +0.7 / +0.1 / +2.1 / +0.4 mAP), which is expected because CGSA was specifically designed for the source-free regime, where no source images are available and thus alignment must be performed purely within the target domain using pseudo labels. In the source-available case, we do not introduce any additional cross-domain alignment losses between source and target features, so the extra supervision from source data is not fully exploited yet.
> > >
> > > Nevertheless, the source-available results remain strong and competitive compared to existing DAOD methods, despite CGSA not being tailored to this easier setting. This suggests that our object-centric design is generally effective for domain adaptation and has significant potential.
> > >
> > > Overall, we would like to express our sincere gratitude to the reviewer for dedicating their time and effort to reviewing our paper. We hope that the above responses have addressed your concerns and provided the necessary clarifications. We are also more than happy to engage in any further discussions.
> > >
> > >
> > > **Reference**
> > > 1. Yao H, Zhao S, Lu S, et al. Source-Free Object Detection with Detection Transformer[J]. IEEE Transactions on Image Processing, 2025.
> > > 2. Zhao Y, Lv W, Xu S, et al. Detrs beat yolos on real-time object detection[C]//Proceedings of the IEEE/CVF conference on computer vision and pattern recognition. 2024: 16965-16974.
> > > 3. Yang J, Li C, Dai X, et al. Focal modulation networks[J]. Advances in Neural Information Processing Systems, 2022, 35: 4203-4217.

---

> > > > ### Author Response · Authors · 2025-11-25
> > > > **A kind reminder from the authors**
> > > >
> > > > Dear Reviewer 5pPM,
> > > >
> > > > Since the discussion period will conclude in slightly more than a week, we would like to ask whether our rebuttal has satisfactorily answered your comments. If there are still aspects that you find unclear or unconvincing, we would very much welcome further discussion and will make every effort to address any remaining concerns. If you now feel comfortable with our responses, we would be very grateful if you could consider raising your score accordingly.
> > > >
> > > > Thank you very much!
> > > >
> > > > The Authors of submission 17563

---

> > > > > ### Comment · Reviewer_5pPM · 2025-11-28
> > > > > **Reply to the authors' feedback**
> > > > >
> > > > > I appreicate the authors' efforts and I am glad that my concerns have been well addressed. I am willing to keep my score. Thanks.

---

> > > > > > ### Author Response · Authors · 2025-11-28
> > > > > >
> > > > > > We sincerely appreciate your positive feedback and thorough review. We will carefully incorporate all the additional content into our revised manuscript. Thank you once again for your insightful comments.

---

### Official Review · Reviewer_jc6a · 2025-10-30

**Soundness:** 2
**Presentation:** 3
**Contribution:** 2
**Rating:** 6
**Confidence:** 4

**Summary:**

This paper proposes a framework called CGSA, specifically designed for source-free domain adaptive object detection (SFA-OD). CGSA embeds object-centric slot representations into DETR queries, progressively disentangles target images into coarse-to-fine slots and aligns slot prototypes with online class prototypes. Experiments show the effectiveness of the proposed CGSA.

**Strengths:**

* The paper is easy to follow.
* Using a slot-aware framework for object-level alignment is a reasonable approach.
* Experiments on five cross-domain shows the effectiveness of the proposed method.

**Weaknesses:**

1. The performance of the base model should be reported(e.g., RT-DETR) for better evaluation.
2. As shown in Figure 3, the HSA module adapts the pre-trained DINO model. Therefore, the additional computation overhead needs to be analyzed.
3. Since the method uses a pre-trained model to inject feature knowledge, it is recommended to add some comparative introductions with existing methods that use VLM for knowledge injection, such as [1][2]. And recent DETR-based SFOD methods should also be discussed, such as [3].

[1] Da-ada: Learning domain-aware adapter for domain adaptive object detection. NeurIPS 2024

[2] SEEN-DA: SEmantic ENtropy guided Domain-aware Attention for Domain Adaptive Object Detection. CVPR 2025

[3] Source-Free Object Detection with Detection Transformer. IEEE TIP 2025

**Questions:**

Please refer to Weakness.

---

> ### Author Response · Authors · 2025-11-22
> **Reply from the Authors (Part 1)**
>
> ## 1. Reporting the performance of the base detector
>
> Thank you for this meticulous comment. In fact, the base DETR performance is already reported in Figure 4 of the main paper, and we have copied it below for your convenience. The **“Source only”** column corresponds exactly to the base detector trained only on the labeled source domain, without any adaptation module.
>
> **Table R2.1** Ablation on four cross-domain benchmark
>
> | Datasets                   | Source only | + HSA | + HSA&CGSC (CGSA) |
> | ------------------------------------ | :----------: | :----: | :------------------: |
> | Cityscapes → Foggy-Cityscapes |        46.2 |  51.8 |            **53.2** |
> | Cityscapes → BDD100K         |        46.6 |  50.3 |            **53.0** |
> | Sim10K → Cityscapes           |        56.9 |  66.3 |            **67.7** |
> | KITTI → Cityscapes           |        50.4 |  59.7 |            **60.8** |
>
> As shown in Table R2.1, adding HSA delivers the majority of the improvement across all datasets, indicating that infusing structural, slot-based visual priors substantially stabilizes target-domain localization. Building on this, CGSC consistently brings additional gains by aligning slot representations with class semantics in a domain-invariant manner. The improvements are stable across diverse domain shifts, which suggests that two components are complementary, producing stronger source-free adaptation overall.

---

> > ### Author Response · Authors · 2025-11-22
> > **Reply from the Authors (Part 2)**
> >
> > ## 2. Computational overhead of HSA and its relation to DINO
> >
> > Thank you for this insightful comment. First, we would like to clarify how DINO is used in our framework. HSA leverages a frozen DINO model (DINO-ViT-B16) only as a feature extractor during self-supervised pretraining of the slot module. DINO is frozen and never fine-tuned together with the detector during source-free adaptation. We chose DINO because it is trained with a self-supervised objective[1], which is highly compatible with HSA. Both methods aim to capture generic object structures without relying on labels.
> >
> > To address the reviewer’s concern, we report the parameters, per-iteration processing time[2], and mAP with and without HSA on two different base detectors: a very lightweight RT-DETR[3] and a large FocalNet-DINO detector[4]. Results are measured on the Cityscapes → Foggy-Cityscapes setting.
> >
> > **Table R2.2** Ablation on computing analysis
> >
> > | Base Detector        | Component     | Parameters | Time     | mAP |
> > |:-------:             |:-------------:|:----------:|:------:    |:---:|
> > | RT-DETR[3]           | -             | 43M        | 269ms      | 46.2    |
> > |                      | + HSA         | 63M        | 343ms      | 51.8    |
> > |                      | $\Delta$%     | 46.5%      | 27.5%      | 12.1%   |
> > | FocalNet-DINO-DETR[4]| -             | 228M       | 780ms      | 43.8    |
> > |                      | + HSA         | 251M       | 809ms      | 55.0    |
> > |                      | $\Delta$%     | 10.1%      | 3.7%      | 25.6%    |
> >
> > As shown in Table R2.2, we can observe that our method achieves significant performance gains in both detection frameworks. Although the introduction of HSA increases parameters by 46.5% and training time by 27.5% on the lightweight RT-DETR, it is essential to note that RT-DETR is designed as an extremely lightweight DETR variant for real-time detection, comparable to YOLO-style models. Consequently, any non-trivial module inevitably results in a relatively large percentage increase in parameters. In contrast, on the much heavier FocalNet-DINO, the relative overhead is much smaller (10.1% parameters, 3.72% training time), yielding a remarkable +25.6% mAP improvement. This makes the computational cost of HSA negligible. These results demonstrate that HSA introduces a controllable and reasonable computational overhead.

---

> > > ### Author Response · Authors · 2025-11-22
> > > **Reply from the Authors (Part 3)**
> > >
> > > ## 3. Comparison with VLM-based DAOD methods and DETR-based SFOD
> > >
> > > Thank you for this thoughtful comment, which brought to our attention DA-Ada[5], SEEN-DA[6], and FRANCK[2]. At first, we emphasize that our framework is purely vision-based model and differs fundamentally from DA-Ada and SEEN-DA:
> > >
> > > - **Backbone and training regime.** DA-Ada and SEEN-DA are built on RegionCLIP, a vision–language model that is fine-tuned using both source and target data under the standard DAOD setting (source-available). Their adapters are trained jointly with the VLM backbone. In contrast, CGSA operates in the source-free setting and uses a standard vision-only detector (DETR) as the only trainable backbone during adaptation. We only use a frozen self-supervised DINO model to pretrain the HSA module offline. DINO is not fine-tuned, and no text encoder is involved at any stage of adaptation.
> > > - **The use of language signal.** Indeed, numerous studies have demonstrated that incorporating additional language inputs and supervision information can significantly boost visual recognition performance [7][8]. DA-Ada and SEEN-DA explicitly exploit text encoders and class names to inject semantic knowledge into detection features (domain-aware adapters and semantic-entropy guided attention). However, our method injects visual structural priors only: HSA learns slot-based decompositions from images, and CGSC aligns these slots with class prototypes computed from visual features. No textual supervision or language embedding is used.
> > >
> > > To address the reviewer’s suggestion, we have added a comparison table R2.3 and clarified how our method differs conceptually from VLM-based approaches as well as how it relates to recent DETR-based SFOD. For C→F and K→C, both DA-Ada and SEEN-DA achieve higher accuracy than our method. This is mainly because they use the much stronger RegionCLIP backbone, whose source-only result on C→F (52.6 mAP) is already close to our fully adapted CGSA, and because they operate under a source-available setting rather than the stricter source-free regime. In contrast, on S→C, CGSA achieves 67.7 mAP, slightly surpassing both methods despite using no text encoder and having no access to source data during adaptation, which demonstrates the strong potential of our fully visual approach.
> > >
> > > **Table R2.3** Results on different datasets. C→F refers to Cityscapes→Foggy Cityscapes. S→C refers to Sim10K→Cityscapes. K→C refers to KITTI→Cityscapes. “SF” indicates whether the method works in the source-free (✓) or source-available (✗) setting.
> > >
> > > | Methods       |SF           | Base Detector      | C→F | S→C | K→C |
> > > |:-------:      |:-----------:|:----------:|:------:|:---:| :---:|
> > > | DA-Ada[5]     | ✗    | RegionCLIP | 58.5   | 67.3| 66.7 |
> > > | SEEN-DA[6]    | ✗    | RegionCLIP | 57.5   | 66.8| 67.1 |
> > > | FRANCK[2]     | ✓    | DETR       | 54.1   | -   | -    |
> > > | CGSA(Ours)    | ✓    | DETR       | 53.2   | 67.7| 60.8 |
> > >
> > > FRANCK[2] is closer to our work, as it is also a DETR-based method designed specifically for the source-free setting. FRANCK introduces query-centric modules by reweighting objectness, enhancing contrastive representation and distilling query features more reliably. Conceptually, FRANCK treats queries as generic feature tokens and focuses on where to aggregate information for each query, while CGSA introduces an object-centric framwork by decomposing features into a small set of object-like slots and aligning these slots with class prototypes, yielding domain-invariant and class-consistent representations.
> > >
> > > Numerically, on C→F our method reaches 53.2 mAP, which is close to FRANCK’s 54.1 mAP under the same SF setting. FRANCK does not release official code yet, and its experiments are limited to C→F under an identical detector configuration. Furthermore, we would like to note that FRANCK was published online in October 2025, whereas the ICLR submission due date was in September 2025. While we will still cite and discuss FRANCK in the final version of our paper, we hope the reviewers understand that a thorough empirical comparison on all datasets was not feasible at the time of submission, given the timing of FRANCK's release.
> > >
> > > Overall, the reason we only considered vision-only models, rather than VLMs or multimodal models, is that we had limited computational resources, which restricted our ability to explore multiple modalities. However, we acknowledge that your suggestion is highly relevant and valuable. We plan to investigate the integration of multimodal approaches in our future work, exploring the potential benefits of combining visual and language inputs.

---

> > > > ### Author Response · Authors · 2025-11-22
> > > > **Reply from the Authors (Part 4)**
> > > >
> > > > **References**
> > > >
> > > > 1. Caron M, Touvron H, Misra I, et al. Emerging properties in self-supervised vision transformers[C]//Proceedings of the IEEE/CVF international conference on computer vision. 2021: 9650-9660.
> > > > 2. Yao H, Zhao S, Lu S, et al. Source-Free Object Detection with Detection Transformer[J]. IEEE Transactions on Image Processing, 2025.
> > > > 3. Zhao Y, Lv W, Xu S, et al. Detrs beat yolos on real-time object detection[C]//Proceedings of the IEEE/CVF conference on computer vision and pattern recognition. 2024: 16965-16974.
> > > > 4. Yang J, Li C, Dai X, et al. Focal modulation networks[J]. Advances in Neural Information Processing Systems, 2022, 35: 4203-4217.
> > > > 5. Li H, Zhang R, Yao H, et al. Da-ada: Learning domain-aware adapter for domain adaptive object detection[J]. Advances in Neural Information Processing Systems, 2024, 37: 103574-103598.
> > > > 6. Li H, Zhang R, Yao H, et al. SEEN-DA: SEmantic ENtropy guided Domain-aware Attention for Domain Adaptive Object Detection[C]//Proceedings of the Computer Vision and Pattern Recognition Conference. 2025: 25465-25475.
> > > > 7. Radford A, Kim J W, Hallacy C, et al. Learning transferable visual models from natural language supervision[C]//International conference on machine learning. PmLR, 2021: 8748-8763.
> > > > 8. Fu S, Yang Q, Mo Q, et al. Llmdet: Learning strong open-vocabulary object detectors under the supervision of large language models[C]//Proceedings of the Computer Vision and Pattern Recognition Conference. 2025: 14987-14997.

---

> > > > > ### Author Response · Authors · 2025-11-25
> > > > > **A kind reminder from the authors**
> > > > >
> > > > > Dear Reviewer jc6a,
> > > > >
> > > > > With the discussion phase approaching its end in just over a week, we would like to respectfully ask whether our responses have adequately addressed the issues you raised. Should any questions remain, we would be more than happy to continue the discussion and will do our utmost to clarify them. If you feel that your concerns have been resolved, we kindly ask whether you might consider increasing your score to reflect this.
> > > > >
> > > > > Thank you very much!
> > > > >
> > > > > The Authors of submission 17563

---

> > > > > > ### Comment · Reviewer_jc6a · 2025-11-27
> > > > > > **Thanks for the rebuttal**
> > > > > >
> > > > > > Thank you for your detailed responses, which have solved most of my concerns. Therefore, I remain positive about this paper. It would be even better if the authors could incorporate these discussions into the main text.

---

> > > > > > > ### Author Response · Authors · 2025-11-27
> > > > > > > **Response to Reviewer jc6a**
> > > > > > >
> > > > > > > We sincerely appreciate your positive feedback and meticulous review. We will carefully incorporate the additional content from the rebuttal period into the final version.

---

### Official Review · Reviewer_WgAU · 2025-10-31

**Soundness:** 2
**Presentation:** 2
**Contribution:** 2
**Rating:** 2
**Confidence:** 4

**Summary:**

This paper studies the problem of Source-Free Domain Adaptive Object Detection (SF-DAOD), where a detector trained on a labeled source domain is adapted to an unlabeled target domain without accessing source data during adaptation. The authors argue that existing SF-DAOD approaches neglect object-level structural information across domains. To address this issue, they propose a new framework named CGSA, which introduces Object-Centric Learning (OCL) into SF-DAOD through slot-aware adaptation. The framework contains two main components: (1) Hierarchical Slot Awareness (HSA), which disentangles image features into slot representations; and (2) Class-Guided Slot Contrast (CGSC), which enhances semantic consistency and promotes domain-invariant adaptation. Experiments on five cross-domain object detection benchmarks show that CGSA achieves consistent performance gains compared to prior SF-DAOD methods. The paper also provides some theoretical analysis to justify the generalization of the proposed approach.

**Strengths:**

The authors conduct experiments on multiple benchmark datasets, showing that their method achieves better performance than previous approaches.

The paper includes theoretical discussions supporting the generalization ability of the proposed method.

**Weaknesses:**

The paper’s description of the proposed method, especially the role of slot attention, is unclear. From the current presentation, it appears that slot attention is applied to the queries in DETR. However, it is not clear how this differs in essence from standard attention mechanisms. The authors should provide a clear comparison or ablation study to demonstrate why slot attention is necessary in this context.

The paper claims that slot attention helps capture object-level features. However, the provided visualization (Figure G.1 in the supplementary materials) does not clearly show the effectiveness of slot-based decomposition. The authors should include more convincing visual evidence or quantitative analysis to illustrate how slot representations correspond to distinct object structures.

The proposed method is built on DETR, while several compared SF-DAOD methods are based on Faster R-CNN. Since DETR generally outperforms Faster R-CNN even without adaptation, the comparison may not be fully fair. The authors are encouraged to implement their framework on Faster R-CNN to ensure a fair comparison and isolate the improvement brought by the proposed adaptation mechanism from that of the detector architecture itself.

**Questions:**

Please see the weakness.

---

> ### Author Response · Authors · 2025-11-22
> **Reply from the Authors (Part 1)**
>
> ## 1. Clarifying the role of slot attention, and why it is necessary.
>
> Thank you for this comment regarding slot attention. We would like to clarify that slot attention is not applied to the **DETR object queries** in our proposed CGSA framework. Instead, it operates on the backbone feature map before the decoder, and produces a small set of slot vectors that are later injected into the queries.
>
> Concretely, given a spatial feature map $ F \in \mathbb{R}^{H \times W \times C} $, we treat the flattened features $ \[ x _ i \] _ {i=1}^{HW} $ as inputs and maintain a fixed number $ K $ of learnable **slots** $ \[ s_k \] _ {k=1}^{K} $. Each iteration includes following four steps:
>
> - ​slots act as **queries**, while spatial features act as **keys/values**;
> - each slot aggregates information from all spatial locations via attention;
> - a competition among slots is enforced by a softmax over the **slot dimension**, so that each pixel is softly assigned to one or a few slots;
> - slots are updated by a recurrent update (GRU + MLP) and trained jointly with a reconstruction loss, which encourages each slot to represent an object-level region.
>
> After several iterations, we obtain $ 𝐾 $ slot vectors summarizing the image at a region-level. These slots are then linearly projected and fused into the DETR object queries through addition. Therefore, slot attention acts as an object-centric bottleneck and prior **before** interacting with the decoder.
>
> To provide a technical comparison, we briefly outline the standard attention mechanisms used in DETR:
>
> - Self-attention in the encoder: every token attends to every other token, but the number of tokens remains $ HW $, and there is no explicit grouping or bottleneck.
> - Cross-attention in the decoder: a fixed set of object queries attends to the encoder tokens, but the queries are free learnable vectors and are not constrained by any reconstruction or competitive assignment among spatial locations.
>
> Therefore, standard attention mixes existing tokens but does not change their organization, and it lacks (i) the competitive assignment of pixels to a limited set of slots, and (ii) the reconstruction objective that pushes each slot to correspond to a structured region. Notably, these two properties are precisely the foundations of our theoretical analysis, which revolves around slot-based structural priors and the variance reduction of background noise.
>
> On the other hand, we emphasize that the performance gains are not merely attributed to "just adding more attention layers". To verify this, we introduce a Transformer-block baseline that replaces HSA while keeping the overall data flow and capacity comparable. We compare the following three variants on Cityscapes → Foggy-Cityscapes.
>
> **Table R1.1** Comparison with a standard Transformer block. “w/ Transformer Block” means replacing HSA with a standard Transformer block.
>
> | Method | mAP|
> | ---| --- |
> | Source Only| 46.2 |
> | w/ Transformer Block | 47.8 |
> | CGSA (Ours)| **53.2**|
>
> The Transformer Block baseline works as follows:
>
> - We apply one standard Transformer encoder block (multi-head self-attention + FFN) on the flattened backbone features, with the same hidden size and comparable parameter count to HSA.
> - We follow the decoder design by initializing $K$ learnable queries and computing cross-attention between these queries and the output features from the encoder, resulting in $K$ vectors.
> -  These $K$ vectors are linearly projected and fused into the object queries using the exact same fusion as in our approach.
>
> From the results in the table R1.1, we can see that replacing a standard Transformer block on top of the framwork yields only a +1.6 mAP improvement. **This small gain is consistent with the intuition that simply increasing capacity and mixing features slightly strengthens the detector, but does not fundamentally improve cross-domain adaptation.** In contrast, **our full CGSA framework achieves 53.2 mAP, which is **+7.0** mAP over Source Only and **+5.4** mAP over the Transformer-block variant. This gap is four times larger than the gain from the extra attention block**.
>
> These results clearly indicate that standard attention is not sufficient to provide the object-centric priors needed for effective SF-DAOD. If the benefit came merely from additional attention capacity, the Transformer block should close most of the gap, which it does not. Instead, the substantial improvement brought by CGSA stems from its slot-based decomposition with reconstruction and competitive assignment, coupled with class-guided contrastive alignment. By forcing the model to represent scenes using a small number of object-level slots, HSA effectively obtains fine-grained domain-invariant feature priors and results in a more robust adaptation process, an effect that a vanilla Transformer block is unable to replicate.

---

> > ### Author Response · Authors · 2025-11-22
> > **Reply from the Authors (Part 2)**
> >
> > ## 2. Visual and quantitative evidence for slot-based decomposition
> >
> > Thank you for your thoughtful comment. In the revised version, we include additional visualizations of the slot masks produced by HSA (see Figure H.1 in Appendix H), where each color corresponds to one slot.
> >
> > Across diverse driving scenes in the BDD100k, the slots consistently partition the image into spatially coherent regions that roughly align with objects or large structures (e.g., cars, road surfaces, buildings, sky, vegetation). **Importantly, this behavior emerges without any pixel-level supervision: HSA is trained purely with a reconstruction objective and competitive assignment among slots.** As a result, the decomposition is not intended to match semantic segmentation perfectly. This is main reason why some regions can span multiple semantic categories, and some small objects may be merged with nearby background. This phenomenon is expected in unsupervised slot-based models and has also been observed in prior object-centric learning works [1][2].
> >
> > What HSA provides, therefore, is not a semantic segmentation, but a rich structural prior: the model is encouraged to represent the scene using a small number of object-like regions, whose boundaries tend to follow strong edges and instance contours. This prior is exactly what our theoretical analysis exploits to reduce background variance and extract domain-invariant features. To further enhance the semantic consistency of these structural slots, we introduce the Class-Guided Slot Contrast (CGSC) module, which aligns slot prototypes with class prototypes during adaptation. HSA focuses on how the scene is decomposed, while CGSC focuses on which class each region should correspond to.
> >
> > While visualizations help interpret the behavior of slots, we agree that the effectiveness of HSA should be primarily judged from quantitative results. In the main paper (Fig. 4), we have already provided ablations isolating the contribution of HSA on four benchmarks. For your convenience, we convert the data into the tabular form shown below.
> >
> > **Table R1.2** Ablation Result of the HSA
> >
> > | Datasets             | Base mAP | +HSA mAP | Gain     |
> > | ------------------------------ | :------------: | :--------: | :--------: |
> > | Cityscapes → Foggy-Cityscapes  | 46.2         | 51.8     | **+5.6** |
> > | Cityscapes → BDD100K           | 46.6         | 50.3     | **+3.7** |
> > | Sim10K → Cityscapes (car only) | 56.9         | 66.3     | **+9.4** |
> > | KITTI → Cityscapes (car only)  | 50.4         | 59.7     | **+9.3** |
> >
> > Across datasets that represent diverse types of domain shift, HSA consistently brings substantial performance gains. As shown in Table R1.2, our method results in strong improvement in each dataset, providing solid quantitative evidence that the slot-based decomposition learned by HSA is broadly effective for SF-DAOD rather than being limited to any specific scenario.
> >
> > Besides, we further observe that the gains are particularly large on the single-class scenarios (Sim10K→Cityscapes and KITTI→Cityscapes). In these settings, the semantic ambiguity across classes is minimal, so the main challenge is to cope with large appearance shifts (e.g., different cameras, backgrounds) while localizing the same “car” category.

---

> > > ### Author Response · Authors · 2025-11-22
> > > **Reply from the Authors (Part 3)**
> > >
> > > ## 3. On the choice of DETR and fairness w.r.t Faster R-CNN baselines
> > >
> > > Thank you for this insightful comment. We agree that detector architecture may influence performance. We would like to highlight that our method is deliberately designed around the DETR architecture and its query tokens. The motivation behind this choice is two-fold:
> > >
> > > * ​**Architectural perspective**​: DETR represents a paradigm shift in object detection by eliminating hand-crafted components such as anchors and non-maximum suppression, providing a more unified and end-to-end learning pipeline. **This design naturally complements our approach, which builds upon the object query mechanism to enable more efficient representation learning**.
> > > * ​**Alignment with recent advances**​: Recent advances in domain adaptive object detection, including FRANCK [8] and TITAN [7], have increasingly adopted DETR-based detectors, underscoring a clear and growing trend within the research community. By aligning with this modern trajectory, our work naturally integrates with contemporary SFOD advancements and facilitates more meaningful comparisons and extensions in future research [7][8].
> > >
> > > Here we provide more details:
> > >
> > > In DETR, a fixed number of object queries form a permutation-invariant set. Each query is intended to represent one potential object, and the final predictions are obtained via bipartite matching between queries and ground-truth boxes. This set-prediction view naturally matches our object-centric design:
> > >
> > > - HSA first decomposes the global encoder feature map into a small set of slots, each summarizing an object-like region.
> > > - These slots are then injected into the object queries, providing them with structured, object-level visual priors before decoding.
> > > - Our theoretical analysis is built on this slot–query coupling, where each query interacts with a small number of slots that encode object-centric structural priors.
> > >
> > > In contrast, Faster R-CNN represents objects via local RoI features cropped from thousands of region proposals. There is no fixed set of global queries, no one-to-one set prediction, and proposals are heavily tied to anchor design and NMS. Mapping our slots onto this pipeline is inherently awkward:
> > >
> > > - Slots represent globally assigned object-level regions, whereas RoIs are local crops determined by RPN. The only feasible option is to fuse the slots with all RoIs, which largely defeats the purpose of competitive global decomposition.
> > > - The proposal-based pipeline already provides its own object hypotheses. Injecting another global decomposition on top tends to be redundant and has much weaker correspondence with the final classification/regression heads than in DETR, where every prediction must go through query–slot interaction.
> > >
> > > For these reasons, we view CGSA as conceptually DETR-specific: it exploits the global set of object queries and the Hungarian matching structure, which are absent in Faster R-CNN.
> > >
> > > While Faster R-CNN is a well-established detector, recent work in domain adaptive object detection has shown that DETR-based detectors are highly competitive and worth studying in their own right. Several UDA methods are built specifically for detection transformers. SFA[3] firstly aligns encoder/decoder token sequences in Deformable DETR for cross-domain detection. AQT[4] introduces adversarial query transformers that perform adversarial alignment directly on DETR feature tokens. MRT[5] and DATR[6] further design teacher–student retraining and class-wise prototype alignment tailored for DETR-style detectors, achieving strong DAOD performance.
> > >
> > > In the SF-DAOD setting, TITAN[7] and FRANCK[8] both explicitly choose DETR as the base detector and design query-centric mechanisms (query-token adversarial learning, query-fused feature reweighting and distillation) to adapt DETR to new domains. These methods already report state-of-the-art SF-DAOD results on DETR, demonstrating that exploring DETR architectures in this setting is not only fair but also desirable.
> > >
> > > Our work follows this line of research: we adopt DETR as the base detecor but introduce a new object-centric perspective via HSA and CGSC. On top of a strong DETR baseline, our method achieves new SOTA performance among SF-DAOD methods, suggesting that object-centric priors are a promising direction for domain adaptation.
> > >
> > > In summary, we are deeply grateful for your insightful comments, which have further improved this work. We would greatly appreciate it if you could consider revising your score if our responses have successfully addressed your concerns. Additionally, we are more than happy to continue the discussion and explore any further questions you may have, with the goal of continually refining and enhancing our research.

---

> > > > ### Author Response · Authors · 2025-11-22
> > > > **Reply from the Authors (Part 4)**
> > > >
> > > > **Reference**
> > > > 1. Locatello F, Weissenborn D, Unterthiner T, et al. Object-centric learning with slot attention[J]. Advances in neural information processing systems, 2020, 33: 11525-11538.
> > > > 2. Seitzer M, Horn M, Zadaianchuk A, et al. Bridging the gap to real-world object-centric learning[J]. arXiv preprint arXiv:2209.14860, 2022.
> > > > 3. Li X, Chen W, Xie D, et al. A free lunch for unsupervised domain adaptive object detection without source data[C]//Proceedings of the AAAI Conference on Artificial Intelligence. 2021, 35(10): 8474-8481.
> > > > 4. Huang W J, Lu Y L, Lin S Y, et al. AQT: Adversarial Query Transformers for Domain Adaptive Object Detection[C]//IJCAI. 2022: 972-979.
> > > > 5. Zhao Z, Wei S, Chen Q, et al. Masked retraining teacher-student framework for domain adaptive object detection[C]//Proceedings of the IEEE/CVF International Conference on Computer Vision. 2023: 19039-19049.
> > > > 6. Chen L, Han J, Wang Y. Datr: Unsupervised domain adaptive detection transformer with dataset-level adaptation and prototypical alignment[J]. IEEE Transactions on Image Processing, 2025.
> > > > 7. Ashraf T, Bashir J. TITAN: Query-Token based Domain Adaptive Adversarial Learning[J]. arXiv preprint arXiv:2506.21484, 2025.
> > > > 8. Yao H, Zhao S, Lu S, et al. Source-Free Object Detection with Detection Transformer[J]. IEEE Transactions on Image Processing, 2025.

---

> > > > > ### Author Response · Authors · 2025-11-25
> > > > > **A kind reminder from the authors**
> > > > >
> > > > > Dear Reviewer WgAU,
> > > > >
> > > > > As the discussion period will end in a little over a week, we would like to kindly check whether our rebuttal has sufficiently clarified your concerns. If there are still points that you feel are not fully addressed, we would be very grateful for another round of discussion and will do our best to respond in detail. If your concerns have been resolved, could you kindly let us know whether you would be able to increase your score accordingly?
> > > > >
> > > > > Thank you very much!
> > > > >
> > > > > The Authors of submission 17563

---

> > > > > > ### Comment · Reviewer_WgAU · 2025-11-26
> > > > > >
> > > > > > Thank you for your response. The newly added experiments and the visualization of the slot attention module have addressed most of my concerns, and I will consider increasing my score. I have one additional question: since the HSA module is pretrained on the COCO dataset, have the authors explored its generalization ability? For example, if the downstream dataset has a distribution that differs significantly from COCO, does the module still perform well?

---

> > > > > > > ### Author Response · Authors · 2025-11-26
> > > > > > > **Follow-up response by the authors**
> > > > > > >
> > > > > > > Thank you for your thoughtful question on the generalization ability of the COCO-pretrained HSA module. We added **two complementary analyses** to address this point.
> > > > > > >
> > > > > > > We first visualize **the feature distributions of COCO and all datasets used in our experiments** (see Figure K.1 in Appendix K). To ensure a fair comparison, we use the same frozen DINO backbone as in HSA pretraining, we extract global image features and apply t-SNE for dimensionality reduction. As a reference case, we also plot COCO vs Pascal VOC [1] (first row, first column). These two classic datasets are both generic web images with centrally framed, prominent objects, and their t-SNE embeddings significantly overlap, confirming that their distributions are relatively similar.
> > > > > > >
> > > > > > > In contrast, when we compare COCO with Cityscapes, Foggy-Cityscapes, BDD100K, Sim10K, and KITTI, the two feature distributions are clearly separated in every plot:
> > > > > > >
> > > > > > > - Cityscapes / BDD100K / KITTI are captured from on-board cameras in real driving scenes, with egocentric viewpoints, long horizons, and structured road layouts.
> > > > > > > - Sim10K is generated from a driving simulation game engine, with synthetic textures and rendering artifacts.
> > > > > > > - Foggy-Cityscapes is created by adding synthetic fog to Cityscapes, further changing low-level statistics.
> > > > > > >
> > > > > > > **All of these are very different from COCO’s general, static object photographs.** Nevertheless, our COCO-pretrained HSA consistently improves performance across all these benchmarks in the main experiments. This strongly suggests that **the COCO-pretrained HSA generalizes well**, as evidenced by its ability to transfer the object-centric structural prior learned from generic web images to very different driving domains, rather than overfitting to COCO-specific statistics.
> > > > > > >
> > > > > > > Furthermore, we also **quantify how much HSA actually depends on COCO pretraining** on the Cityscapes→Foggy-Cityscapes. The ablation (see Table R1.3) compares models with and without COCO self-supervised pretraining of HSA, while keeping all other settings identical.
> > > > > > >
> > > > > > > **Table R1.3** Ablation Study of COCO pretraining for HSA. "Source Only" indicates that "Target testing without target data training and using only the base detector".
> > > > > > >
> > > > > > > | Setting                   | COCO pretraining for HSA | mAP    |
> > > > > > > | :-----------------------: | :----------------------: | :--------: |
> > > > > > > | Source only               | –                        |  46.2 |
> > > > > > > | w/ HSA                    | ✗                       |  51.2 |
> > > > > > > | w/ HSA & CGSC (CGSA)            | ✗                        |  52.7 |
> > > > > > > | w/ HSA                    | ✓                        |  51.8 |
> > > > > > > | w/ HSA & CGSC (CGSA)            | ✓                        |  53.2 |
> > > > > > >
> > > > > > > From Table R1.3, we observe that even **without COCO pretraining**, adding HSA and CGSC already brings **large gains** over the baseline: HSA alone improves mAP from 46.2 to 51.2 (+5.0), and the full CGSA further raises it to 52.7 (+6.5). This shows that HSA can be learned purely from the in-domain data (source and target) and still provides strong object-centric priors for adaptation. COCO self-supervised pretraining gives an additional but **moderate boost**: the mAP of HSA increases from 51.2 to 51.8 (+0.6), and that of CGSA from 52.7 to 53.2 (+0.5).
> > > > > > >
> > > > > > > In other words, **while COCO pretraining is helpful as a better initialization, the core benefits of our approach do not rely on COCO itself**. HSA remains effective when trained from scratch within our source-free pipeline, and **the COCO-pretrained weights generalize well to downstream datasets whose distributions are substantially different from COCO**, as evidenced by both the distribution visualizations and the strong performance on all benchmarks.
> > > > > > >
> > > > > > > We hope this additional analysis addresses your concern, and we truly appreciate your careful consideration. If any concerns remain not fully resolved, we would be very grateful for the opportunity to continue the discussion and will provide detailed explanations.
> > > > > > >
> > > > > > > **Reference**
> > > > > > > 1. Everingham M, Van Gool L, Williams C K I, et al. The pascal visual object classes (voc) challenge[J]. International journal of computer vision, 2010, 88(2): 303-338.

---

> > > > > > > > ### Author Response · Authors · 2025-11-27
> > > > > > > > **A kind reminder from the authors**
> > > > > > > >
> > > > > > > > Dear Reviewer WgAU,
> > > > > > > >
> > > > > > > > This is a follow-up message from the authors. We are writing to kindly check if your concerns have been adequately addressed. If not, we would greatly appreciate it if you could share any further comments, as the discussion period is coming to a close.
> > > > > > > >
> > > > > > > > We sincerely appreciate your insightful comments and efforts throughout the review process.
> > > > > > > >
> > > > > > > > The Authors of submission 17563

---

> > > > > > > > > ### Comment · Reviewer_WgAU · 2025-11-28
> > > > > > > > >
> > > > > > > > > Thanks for providing additional results to address my question. I have no further questions and will increase my score accordingly.

---

> > > > > > > > > > ### Author Response · Authors · 2025-11-28
> > > > > > > > > >
> > > > > > > > > > We are truly encouraged that our responses have addressed your concerns, and we are deeply grateful for your willingness to revise your rating. Once again, we sincerely appreciate your insightful comments during the review process, which will be carefully incorporated into our revised manuscript.

---

### Author Response · Authors · 2025-12-02
**Closing Remarks**

As the rebuttal period comes to a close, we would like to express our sincere gratitude to all reviewers for their time and effort in reviewing our work and providing valuable feedback.

In the initial reviews, our work was recognized for its contributions, including (i) introducing a novel object-centric, slot-aware framework for Source-Free Domain Adaptation Object Detection (SF-DAOD), (ii) achieving consistent SOTA performance on multiple challenging benchmarks, and (iii) providing a solid theoretical analysis that supports the proposed approach.

During the rebuttal phase, we carefully addressed the reviewers' comments regarding computational efficiency analysis, generalization ability evaluation, and deeper empirical validation of theoretical proofs and visualization results. Specifically, we have provided more detailed technical explanations and additional experiments, which we believe fully address the reviewers' concerns. **Fortunately, our responses have been well received, with Reviewer WgAU (R1) explicitly mentioning that the raised concerns have been resolved and will increase the score, while the remaining three reviewers provided positive recommendations, appreciating our work and suggesting acceptance.**

In summary, this paper proposes the first object-centric, slot-aware framework for SF-DAOD. We believe our work will be of interest to the community because our approach not only provides leading performance for SF-DAOD but also holds promising potential for more general domain adaptation problems, which has implications for many real-world applications.

---

### Public Comment · ~Zichong_Chen1 · 2026-03-28
**The confusion about the source-domain pre-training**

The definition of source-free domain adaptation (SFDA) is to transfer a pre-trained model from the source domain to the target domain. From the perspective of academic papers and code implementation, using the source domain is only for obtaining pre-trained models. In practical applications, the source domain data is no longer accessible, and only the trained model can be used. Therefore, I think that introducing the new module (HSA) during pre-training on the source domain is inappropriate. Because this is only for the purpose of paper processing and violates the SFDA rules of practical applications. Although I am not a reviewer, I still hope the author can provide clarification on this matter.

---

> ### Public Comment · ~Boyang_Dai1 · 2026-06-13
> **Clarification on the Source-Free Protocol and the Role of HSA**
>
> We apologize for our delayed reply, as we did not receive any emails or notifications from OpenReview regarding your comments. We agree that, in some practical deployments, the target-side user may only receive a fixed legacy/off-the-shelf source model and may not be able to modify or retrain the source-side model. This is a stricter constraint that may arise in practical deployment, and we understand that it may cause confusion if SFOD is interpreted as requiring adaptation from an arbitrary frozen source model whose architecture and weights cannot be modified at the source side.
>
> However, our work follows the standard SF-DAOD protocol in academic studies, as introduced by one of the earliest and widely recognized SF-DAOD works [1]: the source side first trains a detector using labeled source data, and the target-domain adaptation stage is then performed without accessing or retaining any source images. Under this protocol, **the source-free constraint applies to the adaptation stage**. In our method, HSA is integrated into the detector during source-domain pretraining, and the resulting source-pretrained detector, including HSA, is then used for target-domain adaptation. **During target adaptation, our method only uses the source-pretrained model and unlabeled target images, without any access to source-domain data, thereby satisfying the standard source-free requirement that source data are unavailable during adaptation**.
>
> To further examine whether HSA merely improves the source-only detector rather than helping source-free adaptation, we have conducted an additional ablation during the rebuttal period. When HSA is used only in source-only training, the performance improves from 46.2 to 47.3 mAP, giving a modest gain of +1.1 mAP. In contrast, when source-free target adaptation is enabled with HSA, the performance increases from 46.2 to 51.8 mAP, yielding a much larger gain of +5.6 mAP. These results indicate that the improvement does not hinge on an extra source-side pretraining advantage. More importantly, our work introduces object-centric structural priors as a new perspective for solving the source-free adaptation task.
>
> We are happy to discuss any further questions you may have.
>
> [1] Li, X., Chen, W., Xie, D., Yang, S., Yuan, P., Pu, S., & Zhuang, Y. (2021, May). A free lunch for unsupervised domain adaptive object detection without source data. In Proceedings of the AAAI conference on artificial intelligence (Vol. 35, No. 10, pp. 8474-8481).

---

> > ### Public Comment · ~Zichong_Chen1 · 2026-07-21
> > **Response to the clarification of the author**
> >
> > Thank you for your clarification.

---

### Meta-Review · Area_Chair_4D9T · 2026-01-02

**Summary:**

The paper initially received one negative and three positive ratings. The concerns are mostly about 1) some technical clarifications, e.g., slot attention usage, base detector, 2) experimental results, e.g., effect of pre-trained model and coco dataset, 3) computational overheads.

**Reviewer Concerns:**

The authors have provided responses in the rebuttal to answer initial concerns from the reviewers. The AC took a close look at the paper, reviews, and the rebuttal. After the rebuttal, the AC finds that most questions are addressed well, especially the further clarification of technical contributions and additional experiments. This also leads to the increased rating from reviewer WgAU. Therefore, the AC agrees with the reviewers' overall feedback and hence recommends the acceptance rating, while strongly encouraging the authors to revise the paper accordingly and release the code for reproducibility.

**Reviewer Scores:**

Reviewer WgAU mentioned raising the original rating from 2, and reviewer jc6a and 5pPM indicated to retain the original positive rating, while one reviewer did not fully participate in the discussion.

---

### Decision · Program_Chairs · 2026-01-26

Accept (Poster)